# MVP-LAM: Learning Action-Centric Latent Action via Cross-Viewpoint Reconstruction

Jung Min Lee [1]  Dohyeok Lee [1]  Seokhun Ju [1]  Taehyun Cho [1]  Jin Woo Koo [1]  Li Zhao [2]  Sangwoo Hong [3]  Jungwoo Lee [1 4]

## Abstract

Latent actions learned from diverse human videos serve as pseudo-labels for vision-language-action (VLA) pretraining, but provide effective supervision only if they remain informative about the underlying ground-truth actions. For effective supervision, latent actions should contain information about the underlying actions even though they are inaccessible. We propose **M**ulti-**V**iew**P**oint **L**atent **A**ction **M**odel (**MVP-LAM**), which learns latent actions that are highly informative about ground-truth actions from multi-view videos. MVP-LAM trains latent actions with a *cross-viewpoint reconstruction* objective, so that a latent action from one view must explain the future in another view, reducing reliance on viewpoint-specific cues. On Bridge V2, MVP-LAM produces more action-centric latent actions, achieving higher mutual information with ground-truth actions and improved action prediction, including under out-of-distribution evaluation. Finally, pretraining VLAs with MVP-LAM latent actions improves downstream manipulation performance on various benchmarks. The code and trained checkpoints are available at https://jm-this.github.io/mvp_lam/.

## 1. Introduction

Collecting real-world robot demonstrations remains a central bottleneck in training generalist policies (McCarthy et al., 2024). Unlike foundation models in other domains, robot learning is constrained by the cost of acquiring action-labeled trajectories, which typically requires human teleoperation. This makes large-scale data collection slow and expensive, motivating learning from video as a promising alternative that exploits abundant human manipulation videos to acquire transferable priors over manipulation-relevant dynamics. A fundamental challenge, however, is that such videos do not provide low-level action labels, preventing standard supervised imitation learning.

To address missing actions, recent methods learn *latent actions*, compact representations of video frame transitions, and use them as pseudo-action labels (Ye et al., 2024; Chen et al., 2024b; Bu et al., 2025; Chen et al., 2025b). A latent action model (LAM) encodes frame-to-frame transitions by reconstructing the next observation. These pseudo-labels have been used to pretrain vision-language-action (VLA) models and to define reusable skills for downstream control. For effective VLA pretraining, the key requirement is that latent actions remain informative about the underlying actions even when ground-truth actions are unavailable. Motivated by this, we define an *action-centric latent action* as one that preserves high mutual information (MI) with the action.

A key obstacle for action-centric latent actions is *exogenous noise*, where visual transitions can be spuriously influenced by factors other than the agent's actions yet still correlate with frame-to-frame changes, e.g., people moving in the background (Misra et al., 2024; Nikulin et al., 2025; Zhang et al., 2025). Among these factors, we focus on viewpoint variation, which entangles camera motion with action-driven transitions. This is especially problematic for human videos, which are often captured from egocentric views with substantial viewpoint variation, causing latent actions to overfit viewpoint-specific cues.

We propose **M**ulti-**V**iew**P**oint **L**atent **A**ction **M**odel (MVP-LAM), which learns discrete latent actions that are highly informative about ground-truth actions. MVP-LAM is trained on multi-view videos with a *cross-viewpoint reconstruction* objective, where a latent action inferred from one view is used to predict the future observation in another view. We find that action-centricity comes from the cross-viewpoint objective itself, not merely from multi-view data, as it discourages encoding viewpoint-specific cues.

[1]Seoul National University, Seoul, South Korea [2]Microsoft Research Asia, Beijing, China [3]Konkuk University, Seoul, South Korea [4]HodooAI Labs, Seoul, South Korea. Correspondence to: Jungwoo Lee <junglee@snu.ac.kr>, Sangwoo Hong <swhong06@konkuk.ac.kr>.

*Proceedings of the 43$^{rd}$ International Conference on Machine Learning*, Seoul, South Korea. PMLR 306, 2026. Copyright 2026 by the author(s).

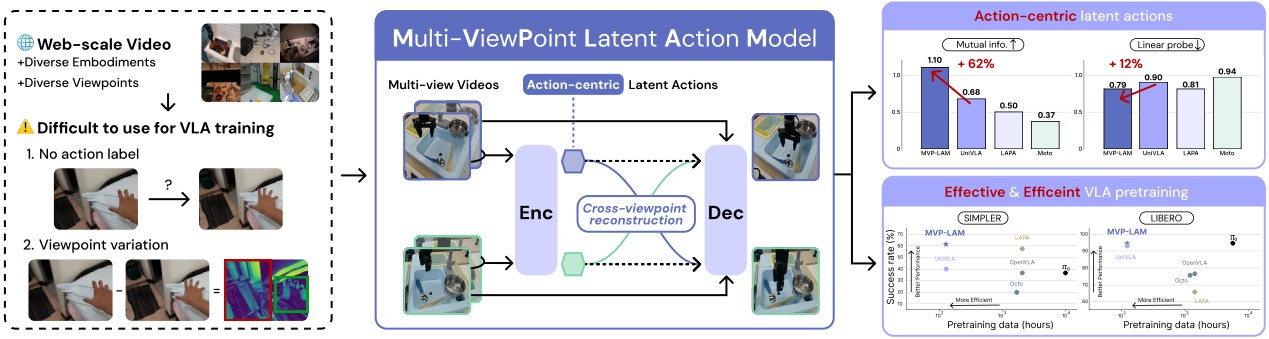

*Figure 1.* **Overview of MVP-LAM.** Web-scale videos lack action labels, and frame-to-frame differences entangle interaction-driven state changes with viewpoint-dependent appearance changes, so identical actions yield different transitions across views while pure viewpoint changes can mimic action-induced ones. MVP-LAM addresses this by encoding the latent from one view and decoding the future frame of a *different* view, removing the incentive to encode view-specific factors and retaining only shared, action-centric information. The resulting latents attain 62% higher mutual information $I(Z; A)$ and 12% lower linear probe error with ground-truth actions than a prior LAM, and enable more effective and data-efficient VLA pretraining, reaching higher success in SIMPLER and LIBERO with an order of magnitude less pretraining video.

Empirically, MVP-LAM learns more action-centric latent actions than LAMs trained on single-viewpoint data with a standard reconstruction objective. On Bridge V2 (Walke et al., 2023) dataset, MVP-LAM achieves higher mutual information between latent actions and ground-truth actions and enables better action prediction accuracy with a simple single linear layer. Also, VLAs pretrained with MVP-LAM latent actions outperform baselines on the SIMPLER (Li et al., 2024) and LIBERO (Liu et al., 2023) benchmarks even with $3\times$ smaller pretraining dataset. Finally, we show that MVP-LAM is robust to real-world noise, including synchronization error, and scalable with the number of viewpoints, dataset ratio, and model size. These results suggest that MVP-LAM can serve as a step toward a *universal latent action model*.

Our contributions are summarized as follows:

1. We introduce MVP-LAM, an unsupervised learning framework that learns latent action well-aligned with ground-truth actions. MVP-LAM is trained on multi-view video dataset with a cross-viewpoint reconstruction objective, where a latent action inferred from one view is used to predict the future observation in another view.

2. We show that MVP-LAM achieves the highest mutual information with ground-truth actions over baselines and improves action prediction on Bridge V2. Moreover, MVP-LAM remains robust to viewpoint perturbations at inference, maintaining consistent transition dynamics across views. This improvement is achieved without action supervision during latent action learning and without relying on the performance of off-the-shelf models.

3. We demonstrate the effectiveness of MVP-LAM latent actions as pseudo-labels for VLA pretraining. VLA pretrained with MVP-LAM outperforms several baselines that use $3\times$ larger pretraining dataset in SIMPLER and LIBERO benchmarks.

## 2. Related Works

**Latent Action Learning from Video.** Recent progress in video-based robot learning has studied how to extract useful representations from large-scale human demonstration videos for downstream control. Several works learn visual priors from videos such as object affordances (Bharadhwaj et al., 2023; Bahl et al., 2023) or trajectory information (Bharadhwaj et al., 2024; Wen et al., 2023). Another line of work learns latent actions as an abstraction of temporal transitions by modeling frame-to-frame visual dynamics without action supervision (Ye et al., 2024; Bruce et al., 2024; Chen et al., 2024b; Bu et al., 2025; Chen et al., 2025a;b; Zhu et al., 2023). Among these works, LAPA (Ye et al., 2024), Moto (Chen et al., 2024b), and UniVLA (Bu et al., 2025) extract latent actions from unlabeled videos and use them as supervision for training downstream embodied AI. In addition, Genie (Bruce et al., 2024), IGOR (Chen et al., 2025a), and AdaWorld (Gao et al., 2025) incorporate latent actions into world models (Ha & Schmidhuber, 2018), improving controllable video generation and supporting downstream embodied planning and manipulation.

Prior latent action approaches study the latent action learning with single-view video, but to our knowledge, none of them explicitly use multi-view video during LAM training. MVP-LAM uses cross-viewpoint reconstruction on multi-view data to construct action-centric latent actions.

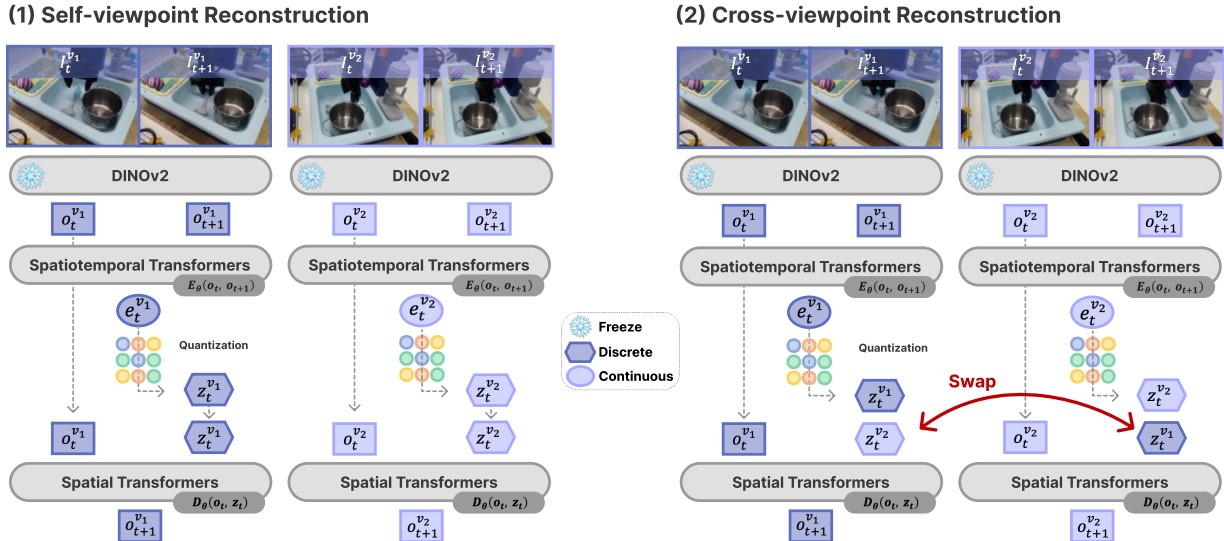

*Figure 2.* **MVP-LAM training with time-synchronized multi-view videos.** (1) *Self-viewpoint reconstruction* (left): for each view $v$, frozen DINOv2 extracts features $(o_t^v, o_{t+1}^v)$. A spatiotemporal encoder produces a continuous latent $e_t^v$ that is vector-quantized into a discrete token $z_t^v$, and a decoder reconstructs $o_{t+1}^v$ from $(o_t^v, z_t^v)$. (2) *Cross-viewpoint reconstruction* (right): MVP-LAM swaps latent tokens across views (e.g., $z_t^{v_1} \leftrightarrow z_t^{v_2}$) while reconstructing each view's future feature, encouraging $z_t$ to capture inherent transition information.

**Learning from Videos with Diverse Viewpoints.** In robot learning, learned policies often exhibit poor generalization across viewpoints due to limited viewpoint diversity in open-source robot datasets (Chen et al., 2024a). One line of work mitigates such limitations via 3D-aware representations (e.g., point cloud) or data augmentation with novel-view synthesis (NVS) models (Driess et al., 2022; Shim et al., 2023; Zhu et al., 2023; Goyal et al., 2023; Ze et al., 2024; Hirose et al., 2022; Tian et al., 2024). Another line of work learns view-invariant representations directly from multi-view data. TCN (Sermanet et al., 2018) uses time-aligned multi-view frames with contrastive learning, while MV-MWM (Seo et al., 2023) and ReViWo (Pang et al., 2025) train multi-view autoencoders to build viewpoint-robust world models for policy learning. However, in-the-wild human manipulation videos often include diverse viewpoints so that they can serve as a scalable source of viewpoint diversity. Accordingly, R3M (Nair et al., 2022) and HRP (Srirama et al., 2024) pretrain visual representations on large-scale egocentric human videos and show improved robustness of downstream policies under viewpoint changes.

These methods primarily aim at observation representations and often require additional components such as camera calibration, dense multi-view coverage of the same scene, or computationally expensive 3D reconstruction and neural rendering. In addition, robustness to viewpoint variation at the level of latent actions has not been widely explored.

**Exogenous Noise in Latent Action Learning.** Exogenous noise in real-world datasets can hinder reliable latent action learning. In the presence of such non-i.i.d. noise, learning representations that include the minimal information necessary to control the agent from videos can require exponentially more samples than learning from action-labeled trajectories (Misra et al., 2024). Theoretically, even linear LAMs tend to capture dominant variation (Zhang et al., 2025), so when noise dominates observation transitions, LAMs are incentivized to encode it rather than the true action (Nikulin et al., 2025). To mitigate this issue, LAOM (Nikulin et al., 2025) incorporates a small amount of action supervision to guide the latent actions. Other approaches reduce the influence of the distractors without action labels, for example, by learning object-centric representations via slot decomposition (Klepach et al., 2025) or by asking vision-language models (VLM) to ignore distractors (Nikulin et al., 2026).

While these methods provide insights for reducing the noise, they introduce additional dependencies, such as action labels, reliable object decomposition, or the quality of pretrained VLMs. In addition, their evaluations are often limited to controlled benchmarks with synthetic distractors (e.g., Distracting Control Suite), leaving open questions about how these methods translate to realistic, noisy manipulation data and whether they yield consistent gains in multi-task or long-horizon settings.

# 3. Method

We propose **MVP-LAM**, a latent action model trained with time-synchronized multi-view videos and a cross-viewpoint reconstruction objective, which produces discrete latent actions as *pseudo-labels* for training VLA models from unlabeled videos.

## 3.1. Problem Formulation

We denote a video by a sequence of images $\{I_t\}_{t=1}^T$. For each timestep $t$, we assume that the image $I_t$ is generated under a camera pose $v_t$. For each image $I_t$, we extract a visual observation in a feature space as $o_t = f(I_t)$, where $f(\cdot)$ is a visual encoder such as DINOv2 (Oquab et al., 2024) or MAE (He et al., 2022). Since video datasets may have different frame rates, we define a fixed temporal stride $H$ and set $o_{t+1} = f(I_{t+H})$.

**Latent action model.** LAM is generally implemented as a vector-quantized variational autoencoder (VQ-VAE) (van den Oord et al., 2017). LAM learns a latent action $z_t$ that summarizes the transition from $o_t$ to $o_{t+1}$. Concretely, an encoder produces a continuous latent $e_t = E_\theta(o_t, o_{t+1})$, which is vector-quantized into a codebook entry, i.e., $z_t = \text{Quantize}(e_t)$. A decoder then predicts the next observation feature as $\hat{o}_{t+1} = D_\theta(o_t, z_t)$. In standard LAM training, the decoder does not take the viewpoint $v_t$ as input. The training objective is

$$\mathcal{L}_\theta(o_t, o_{t+1}) = \|o_{t+1} - \hat{o}_{t+1}\|_2^2 + \mathcal{L}_{\text{quant}} + \mathcal{L}_{\text{commit}}, \quad (1)$$

where $\mathcal{L}_{\text{quant}}$ and $\mathcal{L}_{\text{commit}}$ are the standard VQ-VAE quantization and commitment losses:

$$\mathcal{L}_{\text{quant}} = \|\text{sg}[e_t] - z_t\|_2^2,$$
$$\mathcal{L}_{\text{commit}} = \beta \|e_t - \text{sg}[z_t]\|_2^2$$

where $\text{sg}[\cdot]$ is stop-gradient operator. Since $z_t$ encodes what changes from $o_t$ to $o_{t+1}$, it serves as a discrete representation of the visual transition and can be used as a pseudo-action label when ground-truth actions are unavailable. Furthermore, the discreteness of $z_t$ allows us to train a VLM with a cross-entropy (CE) objective. This pretrained VLM provides an effective initialization for VLA finetuning on downstream tasks.

## 3.2. Action-centric Latent Action

When latent actions are used as pseudo-action labels for behavior cloning policies, it is desirable that the learned latent action $Z_t$ preserves as much information as possible about the underlying action $A_t$.[1] We denote the state by $S_t$, and assume an expert policy induces actions $A_t \sim \pi^\star(\cdot \mid S_t)$

---

[1]We use uppercase letters (e.g., $Z_t$) to denote random variables.

for a given task. In the pretraining stage, we typically do not observe $S_t$ or $A_t$. Instead, we only observe images (or their features) $O_t = f(I_t)$. LAM produces latent actions from consecutive observations, i.e., $Z_t = E_\theta(O_t, O_{t+1})$ (with vector quantization when using VQ-VAE).

Motivated by Zhang et al. (2025), we define a latent action $Z_t$ as *action-centric* if it is highly informative about the underlying action $A_t$. We quantify this by mutual information and consider the objective

$$\max_{Z_t} \mathcal{I}(Z_t; A_t). \quad (2)$$

In this context, viewpoint variation acts as noise. Changes in camera pose $V_t$ can induce frame-to-frame differences in $O_t$ that are predictive of $Z_t$ but are not caused by the action $A_t$. When $Z_t$ is learned under a limited-capacity bottleneck such as vector quantization, allocating representational capacity to viewpoint-dependent factors can come at the expense of action-relevant dynamics and reduce $\mathcal{I}(Z_t; A_t)$. Under simplifying assumptions detailed in Appendix A, one can derive a lower bound

$$\mathcal{I}(Z_t; A_t) \geq \underbrace{\mathcal{H}(Z_t)}_{\text{Total capacity}} - \underbrace{\mathcal{I}(Z_t; V_t, V_{t+1} \mid S_t, S_{t+1})}_{\text{Capacity spent on viewpoint}} - C \quad (3)$$

where $C$ is a constant independent of the latent action $Z_t$. Intuitively, the more capacity $Z_t$ spends on encoding viewpoint information, the less remains for $A_t$. With $\mathcal{H}(Z_t)$ fixed, tightening the bound thus reduces to discouraging viewpoint-dependent variation in $Z_t$.

## 3.3. Multi-Viewpoint Latent Action Model

Building on this motivation, we introduce MVP-LAM, which leverages time-synchronized multi-view videos and cross-viewpoint reconstruction to learn action-centric latent actions. Although single-view capture is easier to collect, multi-view capture remains practical at scale for human videos (Sermanet et al., 2018), with various multi-view human datasets readily available (Kwon et al., 2021; Zheng et al., 2023; Sener et al.; Grauman et al., 2024). For clarity, we describe the two-view case but note that the objective extends to more views.

Given time-synchronized image pairs $\{(I_t^{v_1}, I_t^{v_2})\}_{t=1}^T$, we first extract visual features $o_t^v = f(I_t^v)$ using DINOv2, producing object-centric observation features. For each viewpoint $v \in \{v_1, v_2\}$, the encoder $E_\theta$ predicts a latent action from consecutive observations:

$$e_t^v = E_\theta(o_t^v, o_{t+1}^v), \quad (4)$$
$$z_t^v = \text{Quantize}(e_t^v). \quad (5)$$

As in standard LAMs, the decoder $D_\theta$ is trained to predict the next observation from the current observation and a

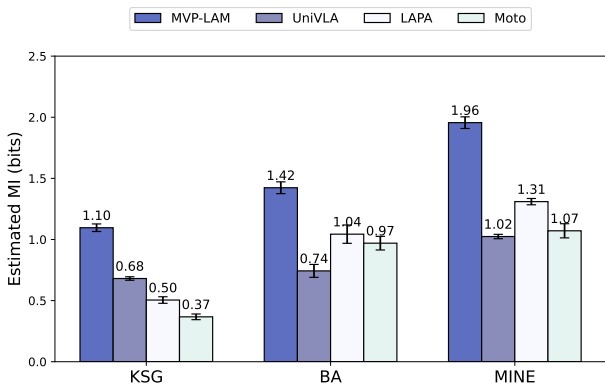

*Figure 3.* **Estimated mutual information.** $\mathcal{I}(Z; A)$ on Bridge V2 with KSG, BA, and MINE estimators. For KSG, latent actions are randomly projected to $d=256$ prior to estimation. Higher is better. Error bars show standard deviation over four seeds.

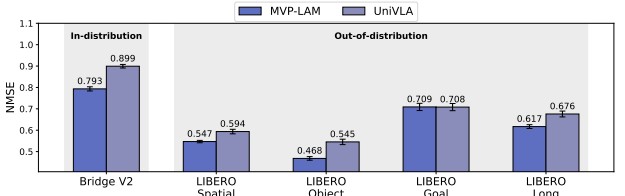

*Figure 4.* **Linear probing result.** NMSE of a linear layer predicting actions from latent actions. Bridge V2 is in-distribution; LIBERO (Spatial/Object/Goal/Long) is out-of-distribution. Lower is better. Error bars show standard deviation over four seeds.

latent action. To reduce the effect of viewpoint variation during LAM training, MVP-LAM optimizes two complementary reconstruction terms: (i) **self-viewpoint reconstruction**, which predicts $o_{t+1}^v$ from $(o_t^v, z_t^v)$ within the same viewpoint, and (ii) **cross-viewpoint reconstruction**, which swaps latent actions across synchronized views and predicts $o_{t+1}^v$ from $(o_t^v, z_t^{\tilde{v}})$ for $v \neq \tilde{v}$. Formally, for two synchronized views $\{v_1, v_2\}$, these terms are defined as

$$\mathcal{L}_{\text{self}} = \frac{1}{2} \sum_{v \in \{v_1, v_2\}} \left\| o_{t+1}^v - D_\theta(o_t^v, z_t^v) \right\|_2^2, \quad (6)$$

$$\mathcal{L}_{\text{cross}} = \frac{1}{2} \sum_{\substack{v, \tilde{v} \in \{v_1, v_2\} \\ v \neq \tilde{v}}} \left\| o_{t+1}^v - D_\theta(o_t^v, z_t^{\tilde{v}}) \right\|_2^2. \quad (7)$$

The full objective of MVP-LAM is

$$\mathcal{L}_{\text{MVP-LAM}} = \mathcal{L}_{\text{self}} + \mathcal{L}_{\text{cross}} + \mathcal{L}_{\text{quant}} + \mathcal{L}_{\text{commit}}. \quad (8)$$

We emphasize that our goal is to learn action-centric latent actions rather than pixel-accurate reconstruction. Even when $\mathcal{L}_{\text{MVP-LAM}}$ cannot be driven to zero under large viewpoint gaps, its gradients steer $Z_t$ toward being action-centric. The full architecture is illustrated in Figure 2.

We now briefly relate cross-viewpoint reconstruction to conditional mutual information in Equation 3. Reducing $\mathcal{L}_{\text{self}}$ and $\mathcal{L}_{\text{cross}}$ enforces $D_\theta(o_t^v, z_t^v) \approx D_\theta(o_t^v, z_t^{\tilde{v}})$ for $v \neq \tilde{v}$. Since the decoder is not conditioned on the viewpoint of the latent action, any viewpoint-specific factors encoded in $z_t^v$ would increase the $\mathcal{L}_{\text{cross}}$. Minimizing $\mathcal{L}_{\text{cross}}$ therefore discourages $z_t^v$ from encoding information that is specific to $(V_t, V_{t+1})$ beyond what is determined by $(S_t, S_{t+1})$. Equivalently, it reduces viewpoint dependence in $Z_t$ and thereby decreases the conditional mutual information $\mathcal{I}(Z_t; V_t, V_{t+1} \mid S_t, S_{t+1})$.

## 4. Experiments

We evaluate whether MVP-LAM learns action-centric discrete latent actions and whether these latent actions serve as effective pseudo-labels for VLA pretraining. Specifically, we address three questions: **RQ1.** Are MVP-LAM latent actions more action-centric? **RQ2.** Do they improve downstream manipulation performance? **RQ3.** Do they preserve transition-relevant information under viewpoint perturbations?

### 4.1. Experiment Setup

**Baselines.** We compare MVP-LAM against the following three representative LAMs. We provide details of the baselines in Appendix D.1.

- **UniVLA** (Bu et al., 2025) learns discrete task-relevant latent action tokens with a VQ bottleneck by encoding consecutive DINOv2 features. We use UniVLA as the primary baseline because MVP-LAM is implemented as a direct modification of UniVLA.

- **LAPA** (Ye et al., 2024) discretizes observation transitions using a VQ-VAE latent action quantizer.

- **Moto** (Chen et al., 2024b) learns a latent motion tokenizer that maps videos to sequences of discrete motion tokens with a large VQ codebook.

**Implementation details.** MVP-LAM follows the UniVLA LAM architecture. For the training dataset, we use time-synchronized multi-view robot trajectories from Open X-Embodiment (OXE) (Collaboration et al., 2023), using the OpenVLA training mixture (Kim et al., 2024), and additionally include multi-view human manipulation videos from EgoExo4D (Grauman et al., 2024). Overall, the training set contains 312k trajectories and we train for 160k steps. The full data mixture and training details of MVP-LAM are provided in Appendix C.1.

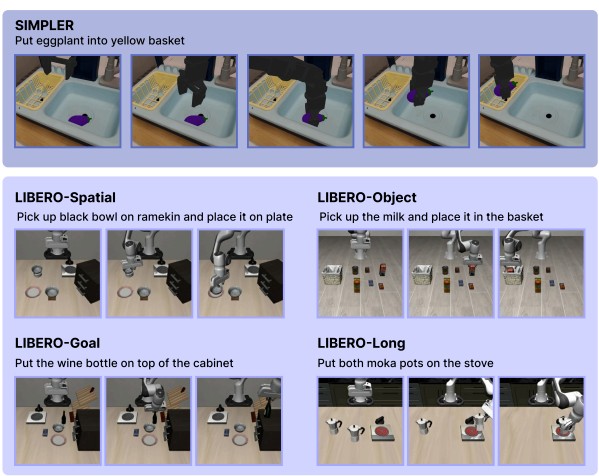

*Figure 5.* **Overview of simulation benchmarks.** Sample observation sequences from SIMPLER and LIBERO suites (Spatial, Object, Goal, and Long) with natural language goal description.

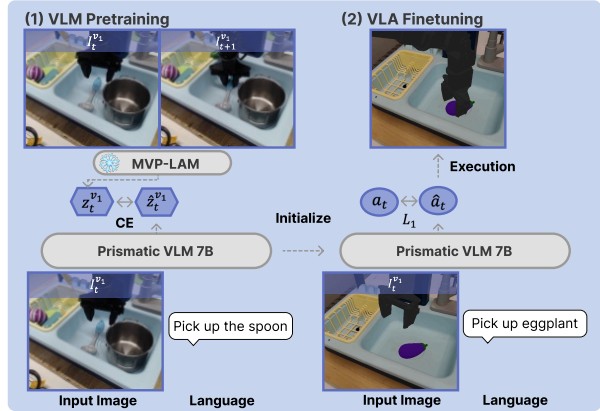

*Figure 6.* **Overview of the VLM pretraining and VLA finetuning.** *(1) VLM Pretraining.* Prismatic-7B VLM is pretrained to predict the discrete latent action token, which is produced by MVP-LAM, from an image and language instruction using a CE loss. *(2) VLA Finetuning.* VLA initializes from the pretrained VLM and finetunes on downstream demonstrations to predict robot actions.

## 4.2. Are MVP-LAM latent actions more action-centric?

We evaluate how action-centric a latent action is by measuring (i) mutual information between latent actions and ground-truth actions, and (ii) how well actions can be predicted from latent actions with a simple linear layer.

**Action normalization across LAMs.** Different LAMs operate at different temporal strides $H$. To make $A_{t:t+H}$ comparable, we convert per-step actions into a *net relative action* over each model's horizon by undoing the dataset-specific normalization, aggregating over the horizon, and renormalizing with original statistics. We provide the details of this process in Appendix B.

**Mutual information estimation.** On Bridge V2, we estimate $\mathcal{I}(Z; A)$ using three estimators: the nonparametric Kraskov–Stögbauer–Grassberger (KSG) estimator, and two variational estimators (Barber–Agakov (BA) (Barber & Agakov, 2003) and a MINE style bound (Belghazi et al., 2018)). We use $k=5$ for KSG. Since KSG is unstable in high dimensions, we apply a random projection to the latent actions so that the overall latent action dimension, including the code length, becomes $d=256$ before KSG. We provide details of MI evaluation in Appendix B.

**Linear probing.** To evaluate the inclusion of ground-truth actions in the latent actions, we use linear probing as Nikulin et al. (2025). Linear probing evaluates how much information is readily accessible in a representation by fitting a simple readout model on top of frozen features (Alain & Bengio, 2017). Here, we freeze the LAM and train a lightweight probe to predict ground-truth actions from latent actions. We use a linear layer $\hat{a}_t = W z_t + b$, where $W$ is the weight

matrix and $b$ is the bias term. We report normalized mean squared error (NMSE), defined as $\mathbb{E}\|a_t - \hat{a}_t\|_2^2/\text{Var}(a)$. To standardize representation dimensionality across methods, we apply PCA to latent actions and keep $d=128$ components, including the code length.

**Minimality of Action-centricity.** Our latent action evaluation metrics measure the action-informativeness of the representation. However, these metrics do not guarantee the *minimality* of the representation, so a latent action that encodes both actions and viewpoints may also exhibit high action-centricity. In this sense, high action-centricity is a necessary but not sufficient condition for a genuinely minimal latent action. To measure the minimality of MVP-LAM, we provide additional evaluation in the Appendix B.2.

**Results and analysis.** As shown in Figure 3, MVP-LAM achieves the highest estimated $\hat{\mathcal{I}}(Z; A)$ across all estimators, suggesting that its latent actions preserve more information about the actions than the baselines. Consistent with MI estimation, Figure 4 shows that MVP-LAM achieves lower NMSE on Bridge V2 and on OOD LIBERO suites (Spatial, Object, and Long), with a small drop on LIBERO-Goal relative to UniVLA. Overall, MI estimation and probing consistently indicate that MVP-LAM learns more action-centric latent actions. We note that UniVLA may struggle to achieve action-centricity because its training objective is primarily driven by task information from language descriptions, which are typically trajectory-level, and this provides weaker supervision for encoding step-level action signals in $z_t$. The details of linear probing and extended analysis, including LAPA and Moto, are listed in Appendix B.

*Table 1.* **SIMPLER benchmark result**. We report success rate and grasping rate (%) on the SIMPLER benchmark. † denotes results reported in prior work. Best is **bolded** and second best is underlined.

| Success Rate | MVP-LAM | UniVLA | LAPA† | OpenVLA† | Octo-Small | Octo-Base | $\pi_0$ |
|---|---|---|---|---|---|---|---|
| StackG2Y | 33.3 | 16.7 | **54.2** | 41.6 | 8.3 | 0.0 | 37.5 |
| Carrot2Plate | **66.7** | 20.8 | 45.8 | 50.0 | 33.3 | 37.5 | 33.3 |
| Spoon2Towel | 66.7 | 54.2 | **70.8** | 37.5 | 25.0 | 12.5 | 29.2 |
| Eggplant2Bask | **75.0** | 66.7 | 58.3 | 16.7 | 12.5 | 20.8 | 45.8 |
| **AVG** | **60.4** | 39.6 | 57.3 | 36.4 | 19.8 | 17.7 | 36.5 |
| **Grasping Rate** | | | | | | | |
| StackG2Y | 54.3 | 45.8 | 62.5 | 50.0 | 54.2 | **70.8** | 58.3 |
| Carrot2Plate | 70.8 | 37.5 | 58.3 | 66.6 | **75.0** | 54.2 | 58.3 |
| Spoon2Towel | 79.2 | 79.2 | **83.3** | 45.8 | 66.7 | 70.8 | 54.2 |
| Eggplant2Bask | 95.8 | **100.0** | 83.3 | 37.5 | 50.0 | 54.2 | 87.5 |
| **AVG** | **75.0** | 65.6 | 71.9 | 50.0 | 61.5 | 62.5 | 64.6 |

## 4.3. Is MVP-LAM Effective for Manipulation?

**Benchmarks.** To examine whether VLA pretrained with MVP-LAM benefits from action-centricity, we evaluate manipulation performance on SIMPLER and LIBERO benchmarks. Figure 5 shows example demonstrations from both benchmarks.

SIMPLER has been shown to correlate with real-world performance even though it is a simulation benchmark. We evaluate four tasks using a 7-DoF WidowX arm to assess generalization across diverse manipulation goals: StackG2Y (stack the green cube on the yellow block), Carrot2Plate (place the carrot on the plate), Spoon2Towel (place the spoon on the towel), and Eggplant2Bask (place the eggplant in the basket). Since SIMPLER does not provide an official finetuning dataset, we use 100 trajectories collected by Ye et al. (2024) (25 per task) and report both grasp rate and success rate.

We further evaluate on four LIBERO suites. LIBERO-Spatial, LIBERO-Object, and LIBERO-Goal evaluate generalization to novel spatial layouts, objects, and goals respectively, and LIBERO-Long evaluates long-horizon manipulation. Each suite contains 10 tasks, and we report the average success rate over 10 rollouts across 50 random seeds.

**Baselines.** We compare VLA pretrained on MVP-LAM latent actions against the following baselines. We provide the implementation details of the baselines in Appendix D.2.

- **Latent action baselines.** UniVLA (Bu et al., 2025) pretrained on Bridge V2 is our primary baseline. It shares the same VLM backbone and the same finetuning and action decoding pipeline, so differences can be attributed to the choice of LAM. In addition, we include LAPA (Ye et al., 2024), which is a representative VLA based on latent actions.

- **VLA baselines.** OpenVLA (Kim et al., 2024) is a VLA model that leverages a large-scale pretraining dataset, including OXE. Octo (Octo Model Team et al., 2023) is transformer-based policy baselines trained on diverse robotic datasets with a unified action representation. Finally, we include $\pi_0$ (Black et al., 2026) which is state-of-the-art VLA model.

**VLA pretraining & finetuning.** Figure 6 shows the details of VLM pretraining and VLA finetuning. We pretrain a VLM to predict MVP-LAM latent actions using a CE objective. We start from a Prismatic-7B VLM checkpoint (Karamcheti et al., 2024) and pretrain on Bridge V2. We then convert the pretrained VLM into a VLA by finetuning with LoRA (Hu et al., 2022) to predict the ground-truth robot action $a_t$. To predict continuous robot action from discrete VLM outputs, we follow the action prediction method of UniVLA based on multi-head attention. Implementation details for VLA pretraining and finetuning are provided in Appendix C.2.

*Table 2.* **LIBERO benchmark results.** Success rate (%) on LIBERO suites for VLAs pretrained on OXE (upper) and Bridge V2 (lower). ∗ indicates methods that use additional wrist-view images and proprioceptive states. Best is **bolded** and second best is underlined.

| Method | Spatial | Object | Goal | Long | AVG |
|---|---|---|---|---|---|
| Octo | 78.9 | 85.7 | 84.6 | 51.1 | 75.1 |
| OpenVLA | 84.7 | 88.4 | 79.2 | 53.7 | 76.5 |
| LAPA | 73.8 | 74.6 | 58.8 | 55.4 | 65.7 |
| $\pi_0*$ | **96.8** | **98.8** | **95.8** | 85.2 | **94.2** |
| UniVLA | 95.2 | 95.4 | 91.9 | 87.5 | 92.5 |
| MVP-LAM | 96.0 | 94.6 | 94.8 | **90.8** | 94.1 |

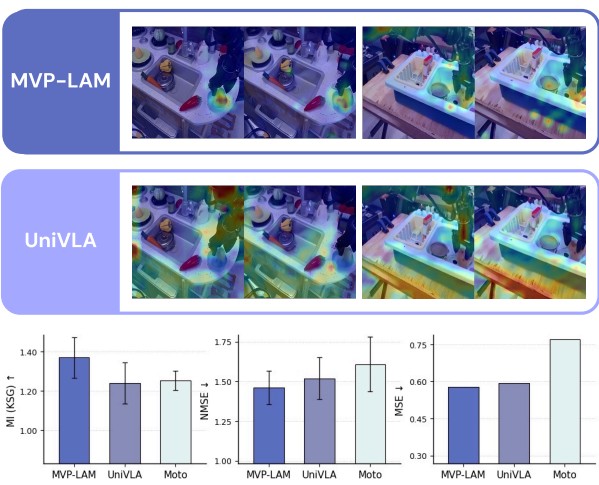

*Figure 7.* **Robustness of latent actions to viewpoint perturbations.** *(Up)* Attention maps on original (left of each pair) and viewpoint-perturbed (right of each pair) transitions for MVP-LAM and UniVLA. *(Down)* Quantitative comparison under viewpoint perturbation. We report MI(KSG) ↑ and linear probe NMSE ↓ from $\tilde{z}_t$ to the ground-truth action $a_t$, and DINOv2-feature reconstruction MSE ↓. Error bars denote standard deviation over 3 seeds.

**Results and analysis.** Table 1 shows SIMPLER results, where pretraining with MVP-LAM's latent actions improves manipulation over other baselines. In particular, MVP-LAM increases the average success rate from 39.6% (UniVLA) to 60.4%, with gains on all four tasks. While LAPA achieves strong performance on some tasks, MVP-LAM remains competitive overall and yields the best average success rate.

Table 2 reports results on LIBERO suites. The VLA pretrained with MVP-LAM achieves 94.1% average success rate, improving over UniVLA under the same Bridge V2 pretraining. Furthermore, despite being pretrained on substantially fewer robot trajectories ($\leq$60k) than OXE ($\geq$970k) and without using LIBERO for either VLM pretraining or LAM training, the VLA pretrained with MVP-LAM outperforms several baselines and remains competitive with the state-of-the-art VLA $\pi_0$, while using only single-view observations, whereas $\pi_0$ additionally relies on wrist-view images and proprioceptive information.

### 4.4. Does MVP-LAM Preserve Transition Information Under Viewpoint Perturbation?

We evaluate whether MVP-LAM preserves transition-relevant information under viewpoint perturbations by using a latent action inferred from a viewpoint-perturbed transition. On Bridge V2, we construct 3.7k viewpoint-perturbed transitions using a novel view synthesis model (Tian et al., 2024). We provide details of viewpoint perturbation in Appendix E.2.

*Table 3.* **Ablations over training data and $\mathcal{L}_{\text{cross}}$.** Robot and Human indicate whether robot or human multi-view videos are included in MVP-LAM training, and $\mathcal{L}_{\text{cross}}$ indicates whether cross-viewpoint reconstruction is enabled. We report NMSE of linear probe and estimated MI (KSG), with $\text{mean}_{\pm\text{std}}$ over 4 seeds.

| Robot | Human | $\mathcal{L}_{\text{cross}}$ | NMSE ↓ | MI (KSG) ↑ |
|:---:|:---:|:---:|:---:|:---:|
| ✓ | | ✓ | $0.91_{\pm0.01}$ | $0.50_{\pm0.03}$ |
| ✓ | ✓ | | $0.96_{\pm0.01}$ | $0.27_{\pm0.01}$ |
| ✓ | ✓ | ✓ | $\mathbf{0.73}_{\pm0.01}$ | $\mathbf{1.10}_{\pm0.03}$ |

**Evaluation setup.** We denote $o_t = f(I_t)$ and $\tilde{o}_t = f(\tilde{I}_t)$ for original image $I_t$ and viewpoint-perturbed image $\tilde{I}_t$. Then, we extract latent actions from the perturbed transitions as $\tilde{z}_t = \text{Quantize}(E_\theta(o_t, \tilde{o}_{t+1}))$. We then follow the same protocols as in Section 4.2 for MI (KSG) and linear probe NMSE, but substitute the perturbed latent $\tilde{z}_t$ to measure robustness under viewpoint perturbation. Both metrics quantify how much information about the ground-truth action $a_t$ is preserved in $\tilde{z}_t$. We additionally report the decoder reconstruction MSE between the predicted next observation $D_\theta(o_t, \tilde{z}_t)$ and the ground-truth $o_{t+1}$ in the DINOv2 feature space, which standardizes the evaluation across models with heterogeneous outputs. For Moto that directly predicts pixels, we embed the decoded frames with DINOv2.

**Results and analysis.** Figure 7 (upper) shows the decoder attention maps of MVP-LAM and UniVLA on the original and viewpoint-perturbed transitions. MVP-LAM concentrates attention on task-relevant regions such as the gripper and the manipulated objects, and remains stable under perturbation, whereas UniVLA's attention is more diffuse and shifts noticeably with the viewpoint change. Figure 7 (lower) reports the quantitative comparison across the three methods. MVP-LAM achieves the highest MI and the lowest linear probe NMSE, indicating that its latent actions retain the most action information under perturbation. On decoder MSE in the DINOv2 feature space, MVP-LAM also attains the lowest error, showing that its robustness does not come at the expense of next-observation prediction. Moto, which decodes pixels, incurs the largest MSE, consistent with their decoders being more sensitive to viewpoint shifts. Additional qualitative results are provided in Appendix E.2.

### 4.5. Ablation Study

We study which components of MVP-LAM are responsible for action-centricity by ablating (i) the human video dataset and (ii) the cross-viewpoint reconstruction. All ablations use the same LAM architecture and training hyperparameters, and follow the same evaluation protocol as Section 4.2.

*Table 4.* **Robustness of MVP-LAM under synchronization error.** Performance under different synchronization lags $\ell$. The results remain consistent regardless of the lag value, indicating robustness to synchronization offsets.

| Metrics | $\ell = 0$ | $\ell = 2$ | $\ell = 4$ |
|---|---|---|---|
| MI (KSG) $\uparrow$ | $1.02_{\pm 0.01}$ | $1.01_{\pm 0.01}$ | $1.02_{\pm 0.02}$ |
| NMSE $\downarrow$ | $0.78_{\pm 0.01}$ | $0.79_{\pm 0.00}$ | $0.80_{\pm 0.01}$ |

**Is the human dataset beneficial to MVP-LAM?** Table 3 shows improved action-centricity on Bridge V2 when human videos are included in MVP-LAM training. In particular, the model trained with human videos outperforms the robot-only baseline on both MI and NMSE. This suggests that including human videos during MVP-LAM training can improve action-centricity. We hypothesize that training MVP-LAM solely on robot data leads to overfitting due to limited motion and scene diversity. LAMs tend to encode factors that explain large frame-to-frame variation in the transitions. Since robot data is collected in relatively controlled settings, the diversity of motion and backgrounds is highly limited, which can increase the risk that the LAM encodes incidental variations in addition to the agent's motion. Meanwhile, human videos provide substantially higher diversity in both motions and scenes, which makes such variations less predictive and encourages the model to prioritize motion as the dominant source of transition, leading to more action-centric latent actions.

**How does cross-viewpoint reconstruction affect?** Table 3 shows that removing $\mathcal{L}_{\text{cross}}$ reduces action-centricity, as reflected by lower MI with ground-truth actions and higher NMSE of MVP-LAM without cross-viewpoint reconstruction. This suggests that training on multi-view videos with $\mathcal{L}_{\text{self}}$ alone is insufficient to learn action-centric latent actions. The observed action-centricity of MVP-LAM is therefore primarily associated with the cross-viewpoint reconstruction, rather than multi-view training alone.

**How robust is MVP-LAM to synchronization error?** Since MVP-LAM relies on paired multi-view videos, it can in principle be vulnerable to synchronization error that arise in practical multi-camera setups due to hardware jitter or asynchronous capture pipelines. To assess robustness under such conditions, we introduce a synthetic lag $\ell$ and form misaligned pairs $(I_t^v, I_{t+\ell}^{\tilde{v}})$, then train MVP-LAM on the Bridge V2 across $\ell \in \{0, 2, 4\}$ frames. Table 4 shows that MVP-LAM remains robust to such synthetic synchronization error, indicating that the method does not require frame-perfect multi-view alignment and is therefore applicable to in-the-wild multi-view datasets where exact synchronization cannot be guaranteed.

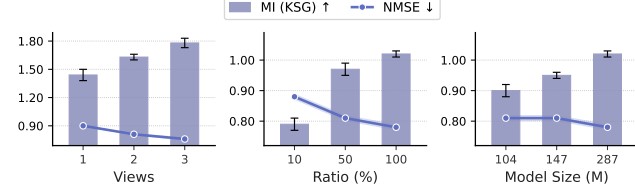

*Figure 8.* **Scaling effect of MVP-LAM.** Action-centricity of MVP-LAM as a function of the number of viewpoints (left), dataset ratio (center), and model size (right). MI $\uparrow$ and NMSE $\downarrow$ consistently improve as each factor increases, demonstrating scaling behavior across all three axes. Shaded area and error bar denotes standard deviation across 4 seeds.

**Scaling Effects of MVP-LAM.** Since MVP-LAM is motivated by dataset scaling, it is important to validate the scalability of MVP-LAM. We consider three scaling axes: (1) the number of viewpoints, (2) the dataset ratio, and (3) the model size. Figure 8 shows that action-centricity of MVP-LAM consistently improves as each factor increases. Notably, increasing the number of viewpoints yields the largest improvement, suggesting that viewpoint diversity is the key to learning action-centric representations.

## 5. Conclusion and Limitations

**Limitations and future works.** Our approach relies on multi-view videos during LAM training. While multi-view capture can be more feasible for human videos than collecting large-scale robot demonstrations, it still requires additional instrumentation compared to single-view data. In addition, while SIMPLER has been shown to correlate with real-world performance, our evaluation on VLA is limited to simulation and does not include real-world robot experiments. A promising direction for future work is to train MVP-LAM on weakly synchronized or pseudo-paired multi-view videos, thereby relaxing the strict synchronization requirement.

**Conclusion.** We propose MVP-LAM, a latent action model that learns discrete latent actions from multi-view videos via a cross-viewpoint reconstruction objective. Across Bridge V2, MVP-LAM produces more action-centric latent actions, as measured by higher mutual information and lower linear-probe NMSE with respect to ground-truth robot actions. When used as pseudo-labels for VLA pretraining, MVP-LAM latent actions yield consistent gains on SIMPLER and LIBERO while requiring substantially less pretraining data than prior VLAs, and remain robust to viewpoint variation as evaluated on novel-view synthesized samples. Beyond these specific results, our findings suggest that multi-view video is a scalable and widely available source of supervision for action-centric latent action learning that requires no action annotations and integrates naturally into existing embodied AI pipelines.

## Impact Statement

This paper presents work whose goal is to advance the field of Machine Learning. There are many potential societal consequences of our work, none which we feel must be specifically highlighted here.

## Acknowledgements

We appreciate Yerin Kim for providing valuable feedback on our figures. This work is in part supported by the National Research Foundation of Korea (NRF, RS-2024-00451435(20%), RS-2024-00413957(20%)), Institute of Information & communications Technology Planning & Evaluation (IITP, RS-2025-02305453(15%), RS-2025-02273157(15%), RS-2025-25442149(15%) RS-2021-II211343(15%)) grant funded by the Ministry of Science and ICT (MSIT), Institute of New Media and Communications(INMAC), and the BK21 FOUR program of the Education, Artificial Intelligence Graduate School Program (Seoul National University), and Research Program for Future ICT Pioneers, Seoul National University in 2026.

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

## A. Relation of Action-centric Latent Action and Viewpoints

We provide the theoretical motivation of reducing the effect of viewpoint variation for action-centric latent actions. For brevity, we drop the time index and write $(S, S') = (S_t, S_{t+1})$ and $(V, V') = (V_t, V_{t+1})$ (similarly for $(O, O')$). We assume the observation $O$ is a deterministic function of $S, V$, i.e. $O = g(S, V)$. We neglect the noise in pixel-level (e.g., lighting variation and sensor noise) since $O$ is often in feature space of the vision encoder. Then,

$$\mathcal{I}(Z; A) = \mathcal{I}(Z; S, A, S') - \mathcal{I}(Z; S, S'|A)$$
$$\geq \mathcal{I}(Z; S, S') - \mathcal{H}(S, S'|A)$$

where $\mathcal{I}(\cdot \, ; \, \cdot)$ is mutual information and $\mathcal{H}(\cdot)$ is entropy. By the chain rule,

$$\mathcal{I}(Z; S, S') = \mathcal{I}(Z; S, V, S', V') - \mathcal{I}(Z; V, V' \mid S, S'),$$

which implies

$$\mathcal{I}(Z; A) \geq \mathcal{I}(Z; S, V, S', V') - \mathcal{I}(Z; V, V' \mid S, S') - \mathcal{H}(S, S' \mid A). \tag{9}$$

Now consider a fixed-capacity discrete bottleneck (e.g., VQ-VAE with codebook size $K$), where $\mathcal{I}(Z; O, O') \leq \mathcal{H}(Z) \leq \log K$. Since we use deterministic encoder $E$ and assume $O = g(S, V)$,

$$0 = \mathcal{H}(Z|O, O') = \mathcal{H}(Z|S, V, S', V') \tag{10}$$

Therefore,

$$\mathcal{I}(Z; S, V, S', V') = \mathcal{H}(Z) \leq \log K \tag{11}$$

Then (9) implies

$$\mathcal{I}(Z; A) \geq \mathcal{H}(Z) - \mathcal{I}(Z; V, V' \mid S, S') - \mathcal{H}(S, S' \mid A). \tag{12}$$

Since $\mathcal{H}(S, S' \mid A)$ is constant under our assumptions, the only representation-dependent term in the bound is $\mathcal{H}(Z)$ and $\mathcal{I}(Z; V, V' \mid S, S')$. Therefore, minimizing $\mathcal{I}(Z; V, V' \mid S, S')$ is beneficial as long as it does not cause representation collapse, i.e., does not substantially reduce $\mathcal{H}(Z)$ under the fixed-capacity constraint.

## B. Action-centricity Estimation Details

**Action normalization.** Robot actions are often provided in a per-timestep normalized space, where each 7D action $a_t$ is z-scored using dataset-level statistics. In our evaluation, we convert such sequences into a *net relative action* representation that aggregates a multi-step action sequence into a single 7D vector while keeping the scale comparable across different horizons.

Specifically, when the actions are stored as $a^{\text{norm}} \in \mathbb{R}^{B \times H \times 7}$, we first recover actions in the original scale via per-dimension de-normalization,

$$a_t^{\text{raw}} = a_t^{\text{norm}} \odot \sigma + \mu, \tag{13}$$

where $(\mu, \sigma)$ are dataset-specific mean and standard deviation and $\odot$ denotes elementwise multiplication. We then form a net action $a^{\text{net}} \in \mathbb{R}^{B \times 7}$ by summing the first six continuous control dimensions over time and taking the final gripper command as the seventh dimension:

$$a_{1:6}^{\text{net}} = \sum_{t=1}^{H} a_{t,1:6}^{\text{raw}}, \qquad a_7^{\text{net}} = a_{H,7}^{\text{raw}}. \tag{14}$$

Finally, we re-normalize the net action with horizon-aware statistics so that the net action remains in a standardized space:

$$\hat{\mu}_{1:6} = H \mu_{1:6}, \quad \hat{\sigma}_{1:6} = \sqrt{H} \, \sigma_{1:6}, \quad \hat{\mu}_7 = \mu_7, \quad \hat{\sigma}_7 = \sigma_7, \tag{15}$$

$$a^{\text{net-norm}} = \left(a^{\text{net}} - \hat{\mu}\right) \oslash \left(\hat{\sigma} + \epsilon\right), \tag{16}$$

where $\oslash$ is elementwise division and $\epsilon$ is a small constant for numerical stability. We use such normalization protocol in both mutual information estimation and linear probing. This aggregation yields a horizon-consistent 7D target: unlike flattening a $H$-step sequence into a $7H$-dimensional label, it keeps the dimension of neural networks fixed across horizons, enabling fair comparisons without changing the capacity. Unlike averaging, summation preserves the semantics of cumulative control and avoids introducing a horizon-dependent rescaling of the target.

*Table 5.* **Hyperparameters for MI estimation and linear probing.** Hyperparameters related to training (upper) and the model (lower) in neural MI estimation and linear probing.

| Hyperparameters | MI estimation | Linear probing |
|---|---|---|
| Batch Size | 1024 | 512 |
| Epochs/Steps | 8000 steps | 30 epochs |
| Learning Rate | 1e-4/5e-5 (MINE/BA) | 1e-3 |
| Scheduler | – | Cosine |
| Gradient Clip | 1.0 | 0.0 |
| Weight Decay | 1e-5 | 0.0 |
| Hidden Dim. | 1024 | 64 |
| Depth | 4 | 1 |

## B.1. Mutual Information

We evaluate how much information the latent action representation $z_t$ retains about the ground-truth action $a$ on the Bridge V2 dataset. Given an observation pair $(o_t^{(i)}, o_{t+1}^{(i)})$, we compute a latent action $z_t^{(i)} = \mathrm{Quantize}(E(o_t^{(i)}, o_{t+1}^{(i)}))$. We estimate the mutual information $\mathcal{I}(Z; A)$ using three complementary estimators: a non-parametric kNN estimator (KSG) and a neural variational estimator (BA, MINE). As a sanity check, we additionally compute a mismatch score by randomly permuting the pairing between $\{z_t^{(i)}\}$ and $\{a_t^{(i)}\}$ at test time, which significantly decreases the estimated dependence. When training the neural MI estimators, we freeze the LAM and optimize only the estimator network.

**KSG (kNN-based MI).** We apply the Kraskov–Stögbauer–Grassberger (KSG) estimator on the paired samples $\{(z_t^{(i)}, a_t^{(i)})\}_{i=1}^N$. Before estimation, we standardize each dimension of $z$ and $a$ using z-score normalization computed on the evaluation split. Since KSG is unstable in high dimensions, we apply a random projection with $W \sim \mathcal{N}(\mathbf{0}, \mathbf{I})$ to each latent action $z_t^{(i)} \in \mathbb{R}^d$.

$$\tilde{z}_t^{(i)} = W z_t^{(i)} \in \mathbb{R}^{256} \tag{17}$$

Since random projection discards information, the estimated mutual information after projection is a lower bound on the true mutual information in the original latent space. We use $k = 5$ for every evaluation.

**MINE (DV variational lower bound).** We train a critic $T_\theta(z, a)$ using the Donsker–Varadhan (DV) representation:

$$\mathcal{I}(Z; A) \geq \mathbb{E}_{p(z,a)}[T_\theta(z, a)] - \log \mathbb{E}_{p(z)p(a)}[\exp(T_\theta(z, a))]. \tag{18}$$

In practice, we approximate samples from $p(z)p(a)$ by shuffling actions within each minibatch (in-batch product-of-marginals). We report the bound on the held-out test split (in bits), and to reduce variance from shuffling, we average the second term over multiple independent shuffles per minibatch.

**Barber–Agakov (BA) variational estimator.** To complement kNN-based and critic-based estimators, we additionally estimate $\mathcal{I}(Z; A)$ using the Barber–Agakov (BA) variational formulation. Starting from

$$\mathcal{I}(Z; A) = \mathcal{H}(A) - \mathcal{H}(A|Z), \tag{19}$$

we introduce a variational conditional density model $q_\phi(a|z)$ and obtain the lower bound

$$\mathcal{I}(Z; A) \geq \mathcal{H}(A) + \mathbb{E}_{p(z,a)}\big[\log q_\phi(a|z)\big]. \tag{20}$$

In practice, we model $q_\phi(a|z)$ as a conditional diagonal Gaussian with mean predicted by an MLP:

$$q_\phi(a|z) = \mathcal{N}\big(a; \ \mu_\phi(z), \ \mathrm{diag}(\sigma^2)\big), \tag{21}$$

where $\mu_\phi(\cdot)$ is an MLP and $\sigma$ is a global (learned) standard deviation shared across samples. We train $\phi$ by maximum likelihood on a training split using paired samples $\{(z_t^{(i)}, a_t^{(i)})\}_{i=1}^N$. To obtain a plug-in estimate of mutual information in

*Table 6.* **Latent action entropy on the MI evaluation set.** We compute $\hat{\mathcal{H}}(Z)$ from the same latent action samples used for KSG MI estimation. Specifically, we treat each quantized latent action vector as a discrete symbol and report its empirical Shannon entropy (in bits). Reporting $\hat{\mathcal{H}}(Z)$ helps contextualize MI results by showing that the compared models have similar marginal entropy of $Z$.

|  | MVP-LAM | UniVLA | LAPA | Moto |
|---|---|---|---|---|
| $\hat{\mathcal{H}}(Z)$ | $14.16_{\pm 0.00}$ | $13.94_{\pm 0.01}$ | $14.29_{\pm 0.00}$ | $14.28_{\pm 0.00}$ |

bits, we also estimate the marginal term $\mathbb{E}_{p(a)}[\log q(a)]$ using a diagonal Gaussian fitted to the training actions,

$$q(a) = \mathcal{N}\big(a;\ \bar{\mu},\ \mathrm{diag}(\bar{\sigma}^2)\big), \tag{22}$$

and report

$$\widehat{\mathcal{I}}_{\mathrm{BA}} = \frac{1}{\log 2}\big(\mathbb{E}_{p(z,a)}[\log q_\phi(a|z)] - \mathbb{E}_{p(a)}[\log q(a)]\big). \tag{23}$$

We evaluate $\widehat{\mathcal{I}}_{\mathrm{BA}}$ on a held-out test split.

**Protocol and reporting.** For the neural estimators (BA and MINE), we train $s_\theta$ or $T_\theta$ on a training split and select the checkpoint based on a validation split (early stopping), then report the final estimate on a disjoint test split. We repeat evaluation across multiple random seeds (which control data subsampling/splitting and optimization randomness) and report the mean and standard deviation. Since different estimators have different biases and scaling, we interpret estimates *within each estimator* and focus on whether the ranking (ours > baseline) is consistent across estimators. Table 5 shows the hyperparameters used in neural estimators. In addition, we report the empirical entropy $\hat{\mathcal{H}}(Z)$ of each model's latent actions on the same Bridge V2 subset used for MI estimation (Table 6). This quantifies the diversity of the latent action codes and helps rule out the trivial explanation that differences in MI are driven primarily by different marginal entropies of $Z$.

## B.2. Details of Linear Probing

**Training details.** For each dataset, we construct a probing set $\{(z_t^{(i)}, a_t^{(i)})\}_{i=1}^N$ and train a simple linear layer to predict actions from latent actions. We minimize the mean-squared error:

$$\mathcal{L}_{\mathrm{probe}} = \mathbb{E}\left[\left\|a_t^{(i)} - \hat{a}_t^{(i)}\right\|_2^2\right], \qquad \hat{a}_t^{(i)} = W z_t^{(i)} + b. \tag{24}$$

As in MI estimation, we freeze the LAM when training the linear probe. Table 5 summarizes the probing hyperparameters.

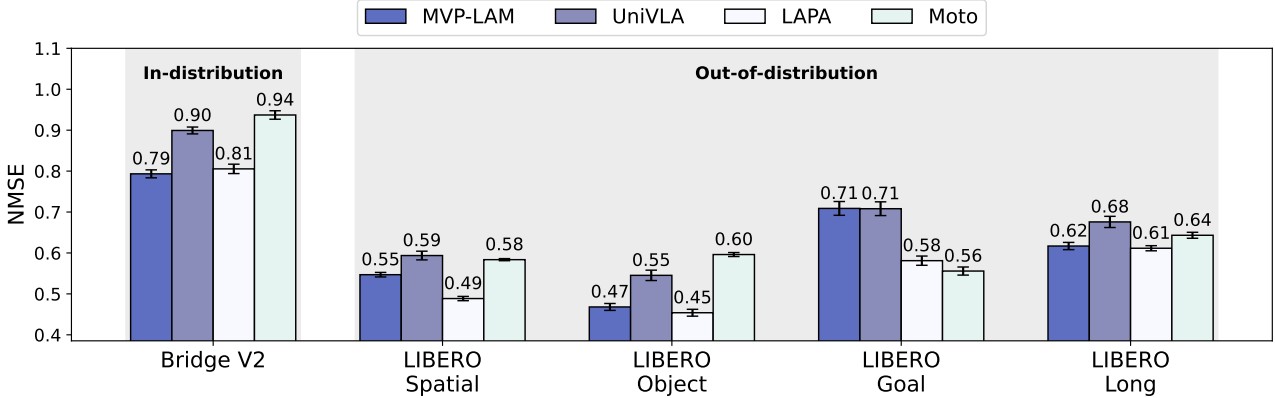

*Figure 9.* **Extended linear probing.** NMSE of a single linear layer predicting normalized actions from latent actions, evaluated in-distribution (Bridge V2) and out-of-distribution (LIBERO suites). Lower is better. Error bars denote standard deviation over four seeds.

**Extended linear probing results.** Figure 9 reports extended linear probing results including LAPA and Moto. Importantly, MVP-LAM achieves the lowest NMSE on Bridge V2 (in-distribution) among *all* compared methods, including LAPA and

Moto, indicating that its latent actions most directly encode step-level robot control signals on the target training distribution. On LIBERO (OOD), LAPA achieves the lowest NMSE on the Spatial, Object, and Long suites, while Moto performs best on LIBERO-Goal. MVP-LAM is second-best on Spatial, Object, and Long, but underperforms on LIBERO-Goal. This pattern indicates that MVP-LAM yields the most action-predictive latents on Bridge V2, while OOD action predictability can be dominated by additional factors that also affect action-centricity beyond viewpoint robustness alone.

We hypothesize why MVP-LAM struggles in LIBERO OOD evaluation: (i) *data scale*: the multi-view robot subset used for MVP-LAM (~55k) is smaller than the training scale used by LAPA (~970k) and Moto (~109k), which can limit generalization in a purely supervised probe; (ii) *token capacity*: LAPA (larger token dim.) and Moto (larger codebook/longer tokens) have higher-capacity bottlenecks, which can capture more action-relevant signals in OOD distribution; and (iii) *viewpoint distribution*: LIBERO is evaluated from a fixed third-person camera, which may better match dominant viewpoints in pretraining corpora used by LAPA and Moto. We expect OOD action predictability to improve by scaling MVP-LAM with larger multi-view robot datasets (e.g., (Khazatsky et al., 2024; AgiBot-World-Contributors et al., 2025)) and additional multi-view human datasets (e.g., (Zheng et al., 2023; Sener et al.; Kwon et al., 2021)) and by increasing bottleneck capacity (larger codebooks and/or higher-dimensional embeddings). Due to the high computational cost of training LAMs at scale, we leave scaling MVP-LAM to larger multi-view datasets and training larger codebooks as future work.

**Minimality of latent actions.**  MI and NMSE of linear probe measure how latent actions include the information about actions, not the minimality of latent actions. Therefore, MI and NMSE would be improved if latent action encodes both actions and the other exogenous noise. To overcome our current evaluation metrics, we conduct inverse linear probing, i.e. predicting latent actions from actions. This evaluation is a proxy of latent action minimality, indicating how action information is included in the latent actions. Table 7 shows that MVP-LAM achieves lower NMSE in Bridge V2 and LIBERO-Long, while underperforms in LIBERO-Goal. This result with Figure 4 suggests that MVP-LAM achieves action-centric latent action which is minimal.

*Table 7.* **Linear probe NMSE of actions to latent actions.** NMSE of latent action to action linear probe in MVP-LAM and UniVLA. Results are reported as $\mathrm{mean}_{\pm\mathrm{std}}$ over 4 random seeds.

| Dataset | MVP-LAM | UniVLA |
|---|---|---|
| Bridge V2 | $\mathbf{0.792}_{\pm\mathbf{0.01}}$ | $0.875_{\pm0.01}$ |
| LIBERO-Goal | $0.643_{\pm0.01}$ | $\mathbf{0.587}_{\pm\mathbf{0.01}}$ |
| LIBERO-Long | $\mathbf{0.631}_{\pm\mathbf{0.01}}$ | $0.650_{\pm0.01}$ |

**Correlation between VLA performance and linear probe.**  Even though a LAM achieves lower NMSE on the linear probe, downstream VLA performance can be affected by various factors, such as the choice of backbone VLM. For instance, even if latent actions were truly identical to ground-truth actions—reducing to the standard VLA setting—performance would still depend on such factors. This may explain why a better linear probe NMSE does not necessarily translate to better VLA performance in Section 4.3. Therefore, we believe that MI and NMSE serve as necessary but not sufficient conditions for improving VLA performance, and they are a kind of diagnostic measures of latent action is used in downstream embodied AI. Jointly optimizing latent action quality with downstream VLA performance is an important future direction.

## C. Details of MVP-LAM

### C.1. MVP-LAM training details

MVP-LAM is trained on a mixture of (i) real-world robot manipulation trajectories and (ii) in-the-wild human manipulation videos. For robot data, we use a subset of Open X-Embodiment (OXE) (Collaboration et al., 2023) that satisfies two conditions: (1) single-arm end-effector control and (2) time-synchronized multi-view trajectories. For human data, we use EgoExo4D (Grauman et al., 2024), which contains ~5k in-the-wild videos with synchronized multi-view recordings.

To match the LfV setting, we do not use proprioceptive inputs or action labels from robot trajectories during MVP-LAM training. Likewise, when using MVP-LAM tokens for VLA pretraining, we only provide visual observations and latent action pseudo-labels. Table 8 lists the datasets and their sampling weights used to train MVP-LAM.

We train MVP-LAM on $4\times$ A6000 GPUs. One epoch takes approximately 96 GPU-hours on $4\times$A6000.

*Table 8.* **MVP-LAM training mixture.** Datasets and sampling weights used for training MVP-LAM.

| MVP-LAM training mixture | |
|---|---|
| Furniture Bench Dataset | 6.58% |
| Taco Play | 7.92% |
| UTAustin Mutex | 6.03% |
| Berkeley Cable Routing | 0.71% |
| Jaco Play | 1.30% |
| Berkeley Autolab UR5 | 3.26% |
| Austin Sirius Dataset | 4.66% |
| Stanford Hydra Dataset | 11.93% |
| IAMLab CMU Pickup Insert | 2.44% |
| NYU Franka Play Dataset | 2.24% |
| Berkeley Fanuc Manipulation | 2.09% |
| Austin Sailor Dataset | 5.88% |
| VIOLA | 2.54% |
| FMB Dataset | 18.94% |
| Austin Buds Dataset | 0.57% |
| Bridge V2 | 14.79% |
| EgoExo4D | 8.12% |

*Table 9.* **Hyperparameters of MVP-LAM.** Details of training (upper) and model architecture (lower).

| Hyperparameters of MVP-LAM | |
|---|---|
| Batch size | 32 |
| Learning rate | $10^{-4}$ |
| Weight Decay | $10^{-2}$ |
| Grad. clip | 1.0 |
| VQ beta | 0.25 |
| Resolution | 224x224 |
| Hidden dim. | 768 |
| Patch size | 14 |
| Num. Blocks | 12 |

## C.2. VLA pretraining and finetuning details

We pretrain a Prismatic-7B VLM (Karamcheti et al., 2024) to predict MVP-LAM latent action tokens with a CE objective, following the UniVLA training recipe. We only use Bridge V2 for VLM pretraining due to limited computational cost. Table 10 summarizes the pretraining hyperparameters. Pretraining is run on $4\times$ H200 GPUs, totaling 45 GPU-hours.

*Table 10.* Hyperparameters used for VLM pretraining with MVP-LAM.

| VLM pretraining hyperparameters | |
|---|---|
| Steps | 200k |
| Learning rate | $2 \times 10^{-5}$ |
| Batch size | 96 |
| Max grad norm | 1.0 |

For finetuning, we follow Bu et al. (2025) and train multi-head attention layers that decode the latent action tokens $z_t$ into continuous robot actions. Specifically, let $o_t = f(I_t)$ and $o_{t+1} = f(I_{t+H})$, and let $(u_v, u_a)$ denote the vision and latent action embeddings from the final layer of the VLM given $o_t$. If the VLM is properly pretrained to predict latent actions, its prediction would be $z_t = \text{Quantize}(E(o_t, o_{t+1}))$. We introduce randomly-initialized, learnable query vectors $q_v$ and $q_a$, and apply multi-head attention as

$$u'_v = \mathcal{A}(q_v, u_v, u_v), \tag{25}$$

$$u'_a = \mathcal{A}(q_a + u'_v, u_a, u_a), \tag{26}$$

$$a_{t:t+H} = \text{MLP}(u'_a) \tag{27}$$

where $\mathcal{A}(Q, K, V)$ denotes a multi-head attention operator with query $Q$, keys $K$, and values $V$. We optimize an $L_1$ regression loss and a CE loss for the token prediction. Table 11 and 12 show the hyperparameters for finetuning in SIMPLER and LIBERO. We finetune the VLA on $2\times$A6000 GPUs, totaling 18 GPU hours for SIMPLER and 30 hours for LIBERO.

*Table 11.* **VLA finetuning hyperparameters on SIMPLER.** We report optimization settings, action decoder hyperparameters, and LoRA configuration.

| VLA finetuning hyperparameters (SIMPLER) | |
|---|---|
| *Training* | |
| Batch size | 4 |
| Gradient accumulation | 4 |
| Steps | 10k |
| *Action decoder* | |
| Learning rate | $10^{-3}$ |
| Weight decay | $10^{-3}$ |
| Window size $H$ | 5 |
| *LoRA* | |
| Rank $r$ | 32 |
| LoRA $\alpha$ | 16 |
| Learning rate | $2 \times 10^{-4}$ |
| Weight decay | 0.0 |

*Table 12.* **VLA finetuning hyperparameters on LIBERO.** We report optimization settings, action decoder hyperparameters, and LoRA configuration.

| VLA finetuning hyperparameters (LIBERO) | |
|---|---|
| *Training* | |
| Batch size | 8 |
| Gradient accumulation | 2 |
| Steps | 30k |
| *Action decoder* | |
| Learning rate | $2 \times 10^{-4}$ |
| Weight decay | $10^{-3}$ |
| Window size $H$ | 12 |
| *LoRA* | |
| Rank $r$ | 32 |
| LoRA $\alpha$ | 16 |
| Learning rate | $5 \times 10^{-5}$ |
| Weight decay | 0.0 |

*Table 13.* **LAM configurations.** $K$ is the codebook size, $L$ is the number of discrete tokens per transition, and $d$ is the token embedding dimension.

| Model | #Codes ($K$) | Code length ($L$) | Code dim. ($d$) |
|---|---|---|---|
| MVP-LAM | 16 | 4 | 128 |
| UniVLA | 16 | 4 | 128 |
| LAPA | 8 | 4 | 1024 |
| Moto | 128 | 8 | 32 |

## D. Additional Baseline Details

### D.1. LAM baselines

Table 13 summarizes the discrete bottleneck configurations used by each latent-action model.

**UniVLA** (Bu et al., 2025) learns *task-relevant* latent actions with a two-stage procedure. In Stage 1, it trains a VQ-VAE LAM with language conditioning to obtain a task-agnostic (task-irrelevant) latent action that explains visual transitions. In Stage 2, it freezes the Stage 1 representation and learns an additional latent action representation that captures the remaining, language-related (task-relevant) information. The resulting discrete tokens are then used as pseudo-action labels for VLA pretraining.

**LAPA** (Ye et al., 2024) is one of the first works to use discrete latent actions as pseudo-action labels for VLA pretraining and demonstrates that such tokens can transfer across embodiments. It learns discrete latent actions via VQ-VAE-style transition tokenization and uses the resulting codes as pseudo-actions during pretraining.

**Moto** (Chen et al., 2024b) learns a motion tokenizer that converts videos into longer sequences of discrete motion tokens. It uses a larger codebook ($K$=128) and longer tokenization ($L$=8) with a smaller per-token embedding dimension ($d$=32), resulting in a higher-capacity token sequence for representing motion.

### D.2. Implementation details of baselines

**Octo.** For both Octo-base and Octo-small, we finetune the language-conditioned policy by updating all parameters (full finetuning) using the official Octo codebase. We finetune for 10k steps with batch size 32 and learning rate of $3 \times 10^{-4}$.

**$\pi_0$.** For SIMPLER finetuning, we finetune $\pi_0$ with LoRA using the official codebase, consistent with the other baselines. We finetune for 10k steps with batch size 16 and learning rate $5 \times 10^{-5}$. For a fair comparison, we finetune using a single RGB image observation and the language instruction, excluding wrist-view images and proprioceptive inputs.

# E. Additional Visualization

## E.1. Latent action examples

Figure 10 visualizes example latent action tokens produced by MVP-LAM for representative frame transitions. We display the discrete codes selected for each transition, along with the corresponding before/after observations. Across examples from different sources, similar motion patterns tend to activate similar codes, illustrating how MVP-LAM clusters transition dynamics in a shared token space without using action supervision.

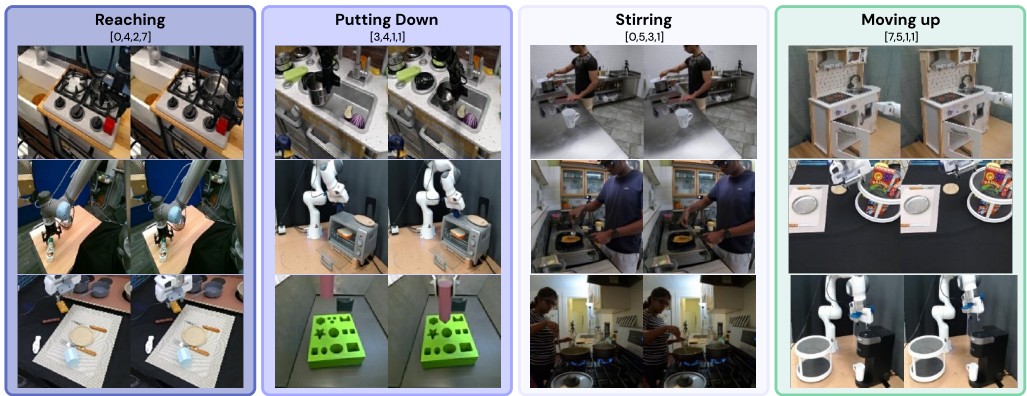

*Figure 10.* **Qualitative latent action visualization.** Example frame transitions and the corresponding MVP-LAM discrete codes selected for each transition.

Figure 11 additionally shows the effect of cross-viewpoint reconstruction objective $\mathcal{L}_{\text{cross}}$. Without $\mathcal{L}_{\text{cross}}$, LAM fails to focus on the manipulation-relevant region while LAM with $\mathcal{L}_{\text{cross}}$ successfully attend the manipulation-relevant region which supports MVP-LAM achieves action-centricity with cross-viewpoint reconstruction.

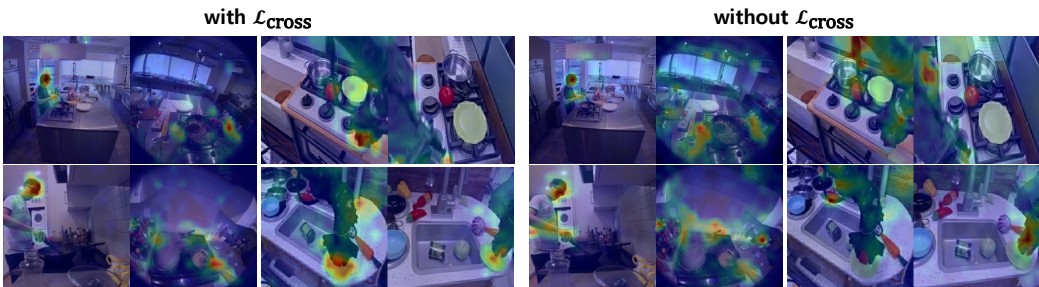

*Figure 11.* **Qualitative comparison of attention with and without cross-viewpoint reconstruction.** Attention maps of MVP-LAM trained with $\mathcal{L}_{\text{cross}}$ (left) and without $\mathcal{L}_{\text{cross}}$ (right). For each sample, we show two different viewpoints of the same state.

## E.2. Details of novel view synthesis in Bridge V2

To evaluate the viewpoint robustness of LAM, we use a zero-shot novel view synthesis (NVS) model finetuned from DROID dataset (Tian et al., 2024). Due to the computational cost of zero-shot novel view synthesis, we use a subset of Bridge V2. We first sample 100 trajectories from Bridge V2 and synthesize 5 perturbed images for each step, totaling 3.7k viewpoint-perturbed transition samples. Given an initial camera pose $(\mathbf{p}_0, \mathbf{q}_0)$, where $\mathbf{p}_0 \in \mathbb{R}^3$ denotes the camera position and $\mathbf{q}_0 \in \mathbb{R}^4$ denotes the camera orientation as a unit quaternion, we sample $N = 5$ perturbed poses by independently applying Gaussian noise to translation and rotation:

$$\Delta\boldsymbol{\theta} \sim \mathcal{N}(\mathbf{0}, \sigma_\theta^2 \mathbf{I}), \qquad \Delta\mathbf{p} \sim \mathcal{N}(\mathbf{0}, \sigma_p^2 \mathbf{I}), \tag{28}$$

where $\Delta\boldsymbol{\theta}$ is a small rotation in axis–angle representation and $\Delta\mathbf{p}$ is a 3D translation. We construct the perturbed pose as $\mathbf{p} = \mathbf{p}_0 + \Delta\mathbf{p}$ and $\mathbf{q} = \Delta\mathbf{q} \otimes \mathbf{q}_0$, where $\Delta\mathbf{q}$ is the unit quaternion converted from $\Delta\boldsymbol{\theta}$ and $\otimes$ denotes quaternion

multiplication. Unless otherwise specified, we use $\sigma_\theta = 0.075$ rad and $\sigma_p = 0.03$ m. We summarize the sampling hyperparameters of the NVS model in Table 14.

*Table 14.* **NVS sampling hyperparameters.** We use DDIM sampling with the following configuration for novel-view synthesis.

| Hyperparameters of NVS model | |
| --- | --- |
| DDIM steps | 250 |
| DDIM $\eta$ | 1.0 |
| Precomputed scale | 0.6 |
| Field of view | 70° |

Figure 12 shows an example of a viewpoint-perturbed trajectory. For each viewpoint perturbation, we randomly sampled the viewpoints within the range where learned perceptual image patch similarity (LPIPS) is smaller than 0.5.

*Figure 12.* **Example of novel view synthesis model in a subset of Bridge V2.** For each step, we synthesize 5 viewpoint-perturbed images with randomly selected viewpoints.

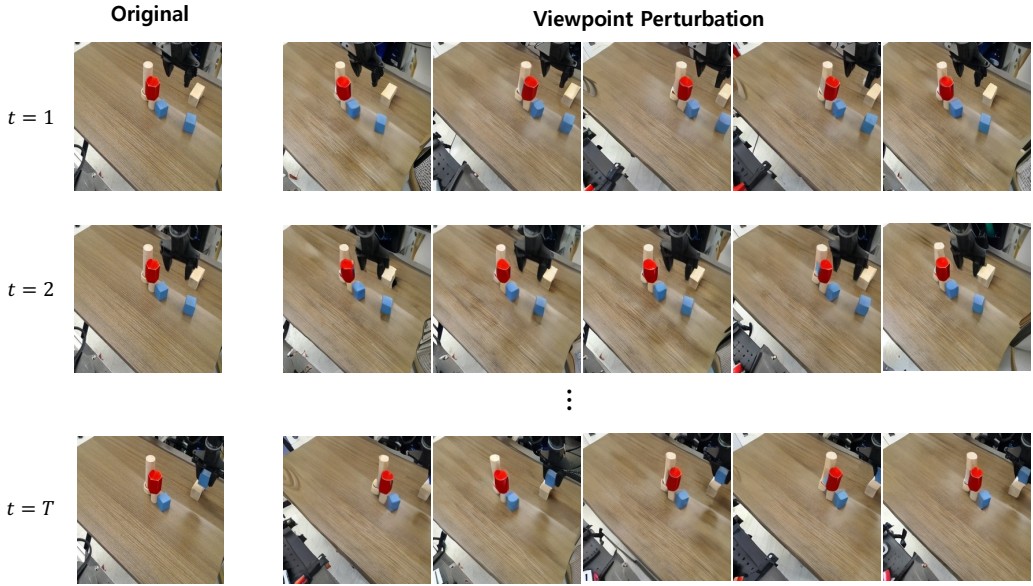

**Additional analysis of viewpoint perturbation of LAPA and Moto**    A potential concern with Figure 7 is that measuring errors in the DINOv2 feature space could disadvantage pixel-decoding LAMs, since their predictions must be re-embedded before computing MSE. To probe this, we additionally evaluate pixel-level reconstruction quality for LAPA and Moto, which explicitly decode RGB frames.

Table 15 reports PSNR on unperturbed transitions and $\widetilde{\text{PSNR}}$ when the latent action is inferred from a viewpoint-perturbed transition. Both methods exhibit a substantial degradation under perturbation, indicating that their failures are already apparent at the pixel level, rather than being an artifact of re-embedding into DINOv2. Qualitative results in Fig. 13 further support this: while predictions remain relatively coherent on the original view, the perturbed setting often produces severely blurred or distorted frames that no longer preserve the scene structure.

This analysis suggests that the higher DINOv2-space errors for pixel-decoding LAMs are consistent with a genuine drop in sample quality under viewpoint-perturbed latent-action inference. At the same time, our models do not decode pixels, so we cannot perform a perfectly symmetric pixel-metric comparison (e.g., PSNR for MVP-LAM). We therefore use DINOv2-space prediction error as a common evaluation space across all methods, and provide the pixel-level results above as supporting evidence that the observed gap is not solely due to the choice of feature-space metric.

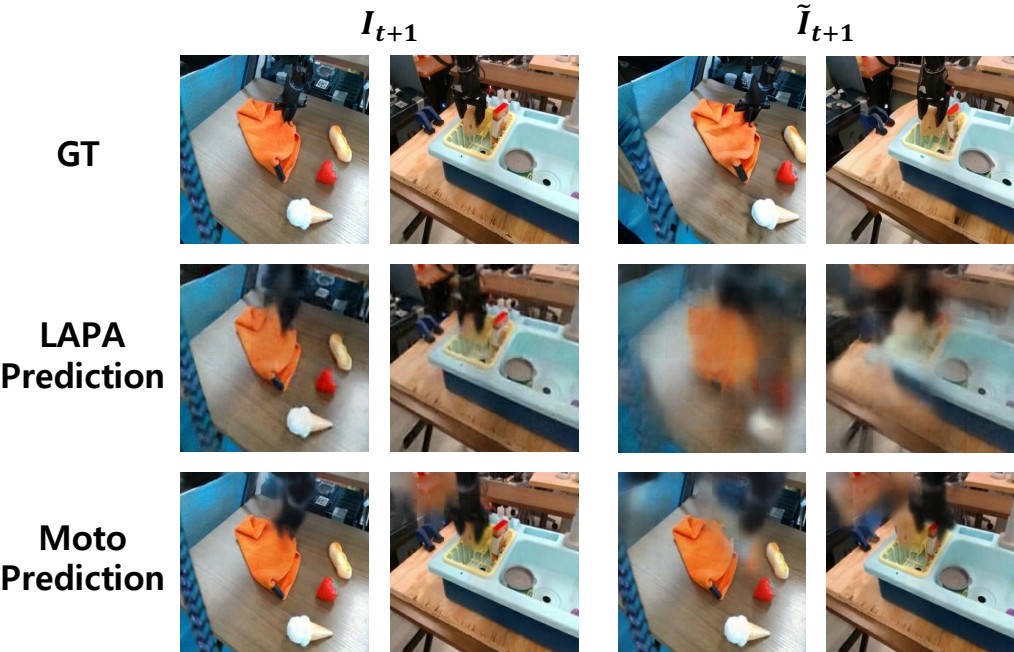

*Figure 13.* **Qualitative reconstructions under viewpoint-perturbed latent actions inference.** Predicted next frames from LAPA and Moto on Bridge V2. While predictions are relatively coherent on unperturbed inputs (left), inferring the latent action from a viewpoint-perturbed transition (right) often leads to visibly degraded reconstructions, consistent with the drop in $\widetilde{\text{PSNR}}$.

*Table 15.* **Pixel-level prediction quality under viewpoint perturbations.** PSNR measures reconstruction quality on unperturbed transitions. $\widetilde{\text{PSNR}}$ measures reconstruction quality when the latent action is inferred from a viewpoint-perturbed transition. Results are reported as mean±std over 3 random seeds.

| Models | PSNR $\uparrow$ | $\widetilde{\text{PSNR}}$ $\uparrow$ |
|---|---|---|
| LAPA | $21.04_{\pm 0.01}$ | $14.91_{\pm 0.01}$ |
| Moto | $23.87_{\pm 0.00}$ | $13.02_{\pm 0.02}$ |

