# OpenReview forum: "MVP-LAM: Learning Action-Centric Latent Action via Cross-Viewpoint Reconstruction"
_ICML.cc/2026/Conference — ICML 2026 regular_

### Official Review · Reviewer_SdiK · 2026-03-09

**Soundness:** 3
**Presentation:** 3
**Significance:** 3
**Originality:** 3
**Overall Recommendation:** 4
**Confidence:** 5

**Summary:**

This paper introduces MVP-LAM, a novel Latent Action Model (LAM) designed to capture view-invariant representations for robotic control. The core innovation lies in the proposed cross-view latent swapping mechanism during training, which encourages the model to decouple view-specific or environmental latent variables from action-relevant information. By ensuring that the latent action remains consistent across different perspectives, the model extracts features more closely aligned with actual physical execution. MVP-LAM demonstrates superior performance over several competitive baselines in experimental evaluations.

**Compliance With Llm Reviewing Policy:**

Affirmed.

**Final Justification:**

This paper proposes a simple yet effective method for training a dynamic representation model or a latent action model. I believe most of the techniques involved are not particularly new in the field of deep learning; therefore, the contribution of this paper is primarily engineering-focused—integrating these techniques into a single model and demonstrating their effectiveness through experiments. While the paper is well-written and free of major logical issues, the level of innovation is not very high. Thus, I will maintain my rating of weak accept.

**Key Questions For Authors:**

How do the authors see the scaling properties of MVP-LAM? The current experiments are somewhat limited in scope. Could you provide additional results or discussions regarding performance across different dataset scales or model sizes? Specifically, do you believe MVP-LAM has the potential to serve as a Universal Latent Action Model if trained on massive, heterogeneous robotic datasets?

**Limitations:**

Yes

**Strengths And Weaknesses:**

### Strengths

1. The paper is well-written, easy to follow, and the proposed methodology is straightforward yet conceptually grounded.

2. The research addresses a highly relevant problem in Embodied AI. Given that modern datasets often provide multi-view observations (e.g., wrist-mounted and static cameras), MVP-LAM is well-positioned for real-world applications. Furthermore, as the field shifts toward utilizing human demonstration data—which often lacks explicit action labels—a robust LAM provides a significant boost to this training paradigm.

3. The approach is elegant and avoids the complexities often associated with contrastive learning. This simplicity likely leads to more stable training and suggests strong potential for scaling up to larger datasets.

---

### Weaknesses

1. The experimental evaluation focuses heavily on internal LAM metrics such as NMSE and Mutual Information (MI). While these are useful diagnostic tools, they do not always correlate perfectly with downstream task performance. The paper would be significantly strengthened by a more thorough discussion and evaluation of how MVP-LAM specifically improves the success rates of VLA models.

2. While the authors discuss human data, the analysis is largely limited to NMSE comparisons. Given that handling unlabeled human video is a primary motivation, it is crucial to demonstrate the downstream impact more directly. Results showing success rates on benchmarks like SIMPLER or LIBERO when using human-pretrained LAMs would provide much more convincing evidence.

3. The discussion on view-invariance is somewhat abstract. The submission would benefit from a qualitative analysis, such as comparing attention maps between models trained with and without the cross-view loss ($L_{cross}$). For instance, visualizing attention on multi-view images from BridgeData could reveal whether $L_{cross}$ effectively shifts the model's focus toward the robot-object interaction and away from viewpoint-specific background clutter.

---

> ### Author Rebuttal · Authors · 2026-03-30
>
> We greatly appreciate the reviewer's detailed and constructive feedback.
> We respond to the main issues below.
>
> > **W1.** Correlation between internal LAM metrics (NMSE, MI) and downstream VLA task performance.
>
> We acknowledge that MI and NMSE alone are not sufficient for predicting downstream VLA performance, since VLA performance is largely affected by how actions are decoded [1] in the finetuning stage.
> However, they serve as necessary diagnostics: methods with poor MI/NMSE consistently fail in downstream tasks, and in our controlled experiments, improved MI/NMSE (MVP-LAM vs. UniVLA) consistently correlates with higher VLA success rates (Tables 1-2).
> Note that even with maximum action-centricity — where the latent action is identical to the ground-truth action, reducing to standard VLA training — downstream performance still depends on many other factors.
> This suggests that improving action-centricity is necessary but not sufficient, and **jointly optimizing latent action quality with downstream VLA performance** is an important future direction.
> To better characterize LAM quality beyond standard NMSE, we additionally conducted bidirectional linear probing (details in **W1**, Reviewer Y3ps), where MVP-LAM outperforms UniVLA in both directions, confirming improvements in both action informativeness and minimality.
>
> > **W2.** Downstream VLA performance with human data pretraining.
>
> We agree that extending the human data ablation (Table 4) to downstream VLA success rates would be more convincing.
> We have confirmed that human data improves action-centricity at the LAM level (Table 4), and as discussed in **W1.**, improved action-centricity is a necessary condition for better downstream performance.
> Whether this improvement fully translates to VLA success rates depends on additional factors beyond LAM quality, which we identified as an important open question in **W1**.
> Although VLA pretraining requires substantial time and computational resources, we appreciate the reviewer's suggestion and will make our best effort to include the results in the revision.
>
> > **W3.** A qualitative analysis, such as comparing attention maps between models trained with and without the cross-view loss.
>
> We appreciate the reviewer for recommending further qualitative analysis.
> We provide additional qualitative analyses at [[this link]](https://q7m4p2x9a1n8.s3.ap-southeast-2.amazonaws.com/f8c1d2.pdf).
> As shown in Figure S2, MVP-LAM generally focuses on robot-object interaction regions, while the baseline trained with only self-viewpoint reconstruction tends to attend to irrelevant factors such as background clutter.
>
> > **Q.** Scaling properties of MVP-LAM across dataset sizes and model sizes, and its potential as a Universal Latent Action Model.
>
> We conduct experiments to examine the scaling behavior of MVP-LAM by ablating the (1) dataset ratio, (2) model size, and (3) token capacity (code length $L$ and the number of codes $K$).
> The following experiments use the Bridge V2 dataset for MVP-LAM training and report MI and NMSE for the dataset ratio and model size ablations and NMSE for the codebook ablation.
> While these experiments use action-centric metrics rather than downstream VLA success rates, the results suggest that MVP-LAM benefits from both larger datasets and higher token capacity.
>
> (1) Dataset ratio
> |Ratio|10%|50%|100%|
> |---|---:|---:|---:|
> |MI $\uparrow$|$0.79\pm0.02$|$0.97\pm0.02$|$1.02\pm0.01$|
> |NMSE $\downarrow$|$0.88\pm0.01$|$0.81\pm0.01$|$0.78\pm0.01$|
>
> (2) Model size
> |Size|104M|147M|287M|
> |---|---:|---:|---:|
> |MI $\uparrow$|$0.90\pm0.02$|$0.95\pm0.01$|$1.02\pm0.01$|
> |NMSE $\downarrow$|$0.81\pm0.01$|$0.81\pm0.01$|$0.78\pm0.01$|
>
> (3) Token capacity
>
> |($L$, $K$)|Bridge|Spatial|Object|Goal|Long|
> |---|---:|---:|---:|---:|---:|
> |(4, 16)|$0.78\pm0.01$|$0.68\pm0.01$|$0.60\pm0.02$|$0.76\pm0.01$|$0.78\pm0.01$|
> |(4, 64)|$0.78\pm0.01$|$0.67\pm0.01$|$0.59\pm0.01$|$0.74\pm0.01$|$0.75\pm0.01$|
> |(8, 16)|$0.77\pm0.01$|$0.63\pm0.02$|$0.50\pm0.02$|$0.75\pm0.01$|$0.73\pm0.00$|
>
> These results suggest that MVP-LAM is scalable with respect to both dataset size and model size, supporting its potential to serve as a universal LAM.
> Furthermore, as discussed in **Q1.** and **Q2.** with Reviewer gaxQ, MVP-LAM demonstrates robustness to synchronization lag and improves consistently with more synchronized views, further supporting its applicability to diverse real-world multi-view settings.
>
> [1] Bu, Q., Yang, Y., Cai, J., Gao, S., Ren, G., Yao, M., Luo, P., & Li, H. (2025). UniVLA: Learning to act anywhere with task-centric latent actions. arXiv. https://arxiv.org/abs/2505.06111

---

> > ### Author Rebuttal · Reviewer_SdiK · 2026-04-03
> >
> > Thank you for the response. The attention maps provided are very promising, and it is encouraging to see that MVP-LAM exhibits more concentrated attention on key objects. I'll maintain my positive rating.

---

> > > ### Author Response · Authors · 2026-04-03
> > >
> > > We sincerely thank the reviewer for the positive feedback. We are glad that the attention maps were helpful in addressing the concerns. We will incorporate them into the revised manuscript.

---

### Official Review · Reviewer_gaxQ · 2026-03-12

**Soundness:** 2
**Presentation:** 2
**Significance:** 2
**Originality:** 2
**Overall Recommendation:** 3
**Confidence:** 3

**Summary:**

This paper proposes MVP-LAM, a discrete latent action model that learns action-centric representations from time-synchronized multi-view videos via a cross-viewpoint reconstruction objective. The core idea is to force the model to predict future observations across different viewpoints using the same latent action, thereby reducing over-reliance on viewpoint-specific cues and emphasizing true agent actions. The method is evaluated on Bridge V2, SIMPLER, and LIBERO-Long benchmarks; results show that MVP-LAM yields latent actions with higher mutual information with ground-truth actions, stronger viewpoint-robustness, and improved performance when used to pretrain vision-language-action (VLA) models for robot manipulation tasks.

**Compliance With Llm Reviewing Policy:**

Affirmed.

**Key Questions For Authors:**

1. You acknowledge the limitation of lacking real-robot validation in Section 5. Do you have plans to conduct real-world experiments in future work? If so, what additional challenges do you anticipate in real-world environments (e.g., lighting variations, dynamic backgrounds, camera calibration errors), and how do you expect MVP-LAM to perform under these conditions?
2. The ablation study only considers two factors: human data and cross-view loss. Important factors such as viewpoint number, synchronization error are not studied. Could the authors supplement more comprehensive ablations?
3. Given that you constrain viewpoint perturbations by an arbitrary LPIPS threshold (< 0.5) . How can we be sure that MVP-LAM’s robustness holds across different perturbation strengths, rather than just within this narrow, arbitrarily chosen range?

**Limitations:**

Yes

**Strengths And Weaknesses:**

Strengths: The paper includes a systematic evaluation pipeline, including mutual information estimation, linear probing, downstream robot manipulation, viewpoint perturbation, and ablation studies. Experimental results generally support the core claims.
Weaknesses: All experiments are limited to simulation environments, with zero real-robot validation. This significantly weakens claims about real-world practicality. Additionally, the method only addresses synthetic viewpoint noise and ignores critical real-world noises such as occlusions, lighting changes, and dynamic backgrounds.

---

> ### Author Rebuttal · Authors · 2026-03-30
>
> We thank the reviewer for the supportive and insightful review.
> We respond to the main issues below.
>
> > **W.** Real-robot validation and robustness to real-world noise beyond synthetic viewpoint perturbations.
>
> We would like to clarify an important distinction: the lack of real-robot validation concerns the downstream VLA, not MVP-LAM itself.
> MVP-LAM is trained and evaluated on real-world dataset including EgoExo4D, which contains in-the-wild human manipulation videos with substantial real-world visual variation covering dynamic backgrounds, lighting changes, and diverse viewpoints.
> The sim-to-real gap is therefore a limitation of the VLA evaluation setup, not of the latent action learning method and its practicality.
>
> Moreoever, our method is **not limited to purely synthetic settings in practice**.
> Our training data already includes human videos that naturally contain real-world visual variation and  we provide additional qualitative analysis at [[this link]](https://q7m4p2x9a1n8.s3.ap-southeast-2.amazonaws.com/f8c1d2.pdf), where Figure S2 demonstrates that MVP-LAM is robust to dynamic backgrounds.
> We also note that the multi-view setting may offer some resilience to occlusions compared to single-view training, as action-relevant information occluded in one view could remain visible in another.
> We will clarify this scope in the revision.
>
> > **Q1.** Additional challenges in real-world environments?
>
> As noted above, MVP-LAM shows its effectiveness even when the real-world noise is included.
> However, we agree that real-world factors especially synchronization error would be critical for MVP-LAM.
> To investigate the impact of synchronization error, we introduced temporal lag in cross-view pairs $(I_t^v, I_{t+\ell}^{\tilde{v}})$ where $\ell \in \{0, 2, 4\}$ and observed that performance is largely maintained even when synthetic lag is present.
>
> | Metric | $\ell=0$ | $\ell=2$ | $\ell=4$ |
> |---|---:|---:|---:|
> | MI $\uparrow$ | $1.02\pm0.01$ | $1.01\pm0.01$ | $1.01\pm0.02$ |
> | NMSE $\downarrow$ | $0.78\pm0.01$ | $0.79\pm0.00$ | $0.80\pm0.01$ |
>
> Overall, the EgoExo4D results provide evidence that MVP-LAM is **useful beyond purely synthetic settings**, while full validation on real-robot downstream control under deployment noise remains future work.
> In the revision, we will make our contributions and limitations more explicit: our contribution is robust latent action learning from real-world multi-view videos, while real-robot downstream validation of VLA pretrained with MVP-LAM remains future work.
>
> > **Q2.** Important factors such as viewpoint number and synchronization error are not studied.
>
> We thank the reviewer for raising this important point.
> As noted above, we provide an ablation on synchronization error.
> We additionally trained MVP-LAM on the Berkeley cable routing dataset with varying numbers of views.
>
> | # of views | 1 | 2 | 3 |
> |---|---:|---:|---:|
> | MI $\uparrow$ | $1.44\pm0.06$ | $1.63\pm0.03$ | $1.78\pm0.05$ |
> | NMSE $\downarrow$ | $0.90\pm0.01$ | $0.81\pm0.01$ | $0.76\pm0.02$ |
>
> Both MI and NMSE improve consistently as the number of viewpoints increases, which implies that using more viewpoints is advantageous to MVP-LAM and its scalability according to the number of viewpoints.
> We will include these results in the revision.
>
> > **Q3.** How can we be sure that MVP-LAM's robustness holds across different perturbation strengths?
>
> We follow the viewpoint augmentation protocol of Tian et al. [1] and **we do not choose the LPIPS threshold in an arbitrary way or in our favor**.
> Tian et al. use an LPIPS threshold to filter out *unrealistic novel-view samples*, and without this filtering, the NVS model can produce clear failure cases such as extreme close-up views caused by poor extrinsics, making it *difficult to determine whether performance degradation stems from the LAM itself or from NVS artifacts*.
> The LPIPS threshold was therefore adopted not to restrict evaluation to a favorable regime, but to ensure that the perturbations remain realistic and that comparisons across models are meaningful.
> Thus, the results of Table 3 and the qualitative analysis in [[this link]](https://q7m4p2x9a1n8.s3.ap-southeast-2.amazonaws.com/f8c1d2.pdf) demonstrate that MVP-LAM is robust to viewpoint perturbations within the valid operating range of the NVS model.
>
> [1] Tian, S., Wulfe, B., Sargent, K., Liu, K., Zakharov, S., Guizilini, V. C., & Wu, J. (2024). View-invariant policy learning via zero-shot novel view synthesis. In 8th Annual Conference on Robot Learning. https://openreview.net/forum?id=tqsQGrmVEu

---

### Official Review · Reviewer_ngzd · 2026-03-13

**Soundness:** 3
**Presentation:** 3
**Significance:** 3
**Originality:** 3
**Overall Recommendation:** 4
**Confidence:** 4

**Summary:**

This paper investigates how to learn discrete latent actions from videos without ground-truth action labels for Vision-Language-Action pretraining. To mitigate the exogenous noise caused by viewpoint variations, the authors propose the Multi-ViewPoint Latent Action Model. This method utilizes time-synchronized multi-view videos and trains with a cross-viewpoint reconstruction objective—forcing the model to predict the future observation of another view using the latent action extracted from the first view. This mechanism prevents the model from encoding viewpoint-specific information into the latent actions, thereby extracting more pure, action-centric features. Experiments on benchmarks such as Bridge V2, SIMPLER, and LIBERO-Long demonstrate that the latent actions generated by MVP-LAM have higher mutual information with ground-truth actions and can significantly improve the fine-tuning success rate of downstream VLA models.

**Compliance With Llm Reviewing Policy:**

Affirmed.

**Final Justification:**

After considering both the paper and the rebuttal, I find my concerns adequately addressed and therefore maintain my positive recommendation.

**Key Questions For Authors:**

1. The strict requirement for time-synchronized multi-view data is the primary scalability bottleneck. Have the authors explored or considered relaxing this to "pseudo-paired" data—for example, using novel view synthesis (NVS) models to generate synthetic alternate views from single-view videos during training? Given that the paper already employs an NVS model for evaluation (Section E.2), repurposing it as a data augmentation strategy during LAM training seems like a natural extension. What are the anticipated challenges (e.g., NVS artifacts leaking into latent actions)?

2. The same decoder $D_\omega$ is used for both self-viewpoint and cross-viewpoint reconstruction (Eq. 6-7). In the cross-view setting, the decoder must predict $o_{t+1}^v$ from $(o_t^v, z_t^{\tilde{v}})$, where the latent action originates from a potentially very different viewpoint (e.g., ego vs. exo in EgoExo4D). How sensitive is reconstruction quality—and consequently latent action purity—to the magnitude of the viewpoint gap between synchronized views? Have the authors measured performance as a function of inter-view angular or translational distance?

3. MVP-LAM underperforms LAPA on LIBERO-Goal in OOD linear probing (Fig. 6), yet achieves state-of-the-art downstream VLA fine-tuning on LIBERO-Long (90.8%, Table 2). The authors attribute the OOD probing gap to data scale (55k vs. 970k), token capacity (code dim 128 vs. 1024), and viewpoint distribution mismatch (Appendix B.2). However, this raises a deeper question: does the NMSE metric of a frozen linear probe actually reflect the feature quality that matters during end-to-end VLA fine-tuning? If not, what alternative intrinsic metrics would the authors recommend for predicting downstream fine-tuning efficacy?

4. Table 6 reports $\hat{H}(Z) \approx 14$ bits against a theoretical maximum of 16 bits ($K=16, L=4$). While overall utilization appears healthy (~88%), have the authors analyzed per-position code distributions? Specifically, is there evidence of partial codebook collapse at certain token positions, and would increasing codebook size $K$ or code length $L$ improve OOD generalization—potentially addressing the capacity concern raised in the LIBERO probing results?

**Limitations:**

yes

**Strengths And Weaknesses:**

### Strengths

1. The paper accurately identifies a core pain point in Learning from Video: exogenous noise like viewpoint variations confounds latent action representation learning. The proposed cross-viewpoint reconstruction is a natural, elegant, and intuitive solution that forces the information bottleneck to filter out viewpoint-specific details and retain genuine state transitions. The theoretical derivation based on mutual information provided in Appendix A offers solid mathematical backing for this intuition.
2. The fine-tuning results on SIMPLER and the long-horizon task LIBERO-Long are impressive. Notably, despite using a relatively small scale of robot data (<60k trajectories) combined with human data, MVP-LAM outperforms baselines like OpenVLA that were pretrained on the massive OXE dataset. This strongly demonstrates the value of high-purity pseudo-action labels for VLA pretraining.
3. The authors evaluate not only downstream success rates but also internal representations via mutual information estimation (KSG, MINE, BA), linear probing, and perturbation tests using Novel View Synthesis (NVS). This rigorous evaluation framework increases confidence in the conclusion that the learned features are indeed action-centric and viewpoint-robust.

### Weaknesses

1. The primary advantage of learning latent actions from videos should be the ability to seamlessly leverage massive in-the-wild single-view videos. However, MVP-LAM strictly relies on time-synchronized multi-view videos. Although the authors utilized datasets like EgoExo4D, acquiring large-scale, time-synchronized, multi-view interaction videos in the real world remains highly costly and difficult, which weakens the method's potential to scale.
2. In the OOD linear probing experiments (LIBERO suites), MVP-LAM's performance in extracting action information lags behind LAPA and Moto on tasks like LIBERO-Goal. This suggests that MVP-LAM might be overfitted to the pretraining distribution (Bridge V2), and its discrete token capacity may be insufficient to maintain strong action prediction capabilities in entirely new scene distributions.
3. The cross-viewpoint reconstruction uses a single shared decoder $D_\omega$ that must predict $o_{t+1}^v$ from $(o_t^v, z_t^{\tilde{v}})$, where the latent action comes from a potentially very different viewpoint (e.g., ego vs. exo in EgoExo4D). The paper does not analyze how decoder performance degrades as the angular or spatial disparity between view pairs increases. If the decoder struggles with large viewpoint gaps, the cross-viewpoint loss gradient signal could become noisy or misleading, potentially undermining the quality of the learned latent actions for extreme view pairs.
4. While SIMPLER simulations correlate with reality, the evaluations are limited to simulation benchmarks and do not include real-world robot experiments.

---

> ### Author Rebuttal · Authors · 2026-03-30
>
> We sincerely appreciate the reviewer's encouraging comments.
> Below, we respond to the main concerns.
>
> > **W1.** Reliance on time-synchronized multi-view videos.
>
> While we acknowledge that time-synchronized multi-view videos introduce a scalability bottleneck, we want to emphasize that our approach remains practical.
> Synchronized multi-view capture is substantially cheaper than obtaining action labels [1], and despite using only 300K multi-view trajectories, MVP-LAM outperforms baselines trained on 970K trajectories, suggesting meaningful data efficiency gains.
> Large-scale multi-view datasets are also increasingly available (Appendix B.2), and extending MVP-LAM to pseudo-paired or synthesized data is a promising direction we discuss in **Q1**.
>
> > **W2.** Overfitting to the pretraining distribution (Bridge V2) and insufficient discrete token capacity to maintain strong action prediction capabilities.
>
> If overfitting to Bridge V2 were the dominant issue, we would expect consistent degradation across all OOD suites.
> Since the gap appears mainly on LIBERO-Goal while MVP-LAM still performs well on LIBERO-Long, the degradation may reflect other factors such as the representation backbone or suite-dependent factors.
>
> > **W3. & Q2.** Sensitivity of decoder performance to viewpoint gap magnitude between synchronized view pairs.
>
> We acknowledge that decoder performance is likely to degrade under very large viewpoint gaps.
> A direct analysis would require viewpoint annotations or calibrated camera which is unavailable in our datasets.
> However, the reconstruction quality of the decoder is not our final goal, but *learning more action-centric latent actions is*.
> Even though MVP-LAM is trained on multi-view datasets with large viewpoint gaps such as EgoExo4D, we found that adding human video data in MVP-LAM training is effective (Table 4).
> This implies that MVP-LAM learns action-centric features from human video datasets even under large viewpoint gaps, *suggesting that the cross-viewpoint reconstruction loss provides useful gradients rather than misleading ones*.
>
> > **Q1.** Relaxing to pseudo-paired data?
>
> Both finetuned NVS models [2] and pseudo-pairing protocols [3] offer promising directions for generating multi-view data from single-view videos.
> However, even though using pseudo-paired datasets is a natural extension for MVP-LAM, we focus on how to exclude the exogenous noise from viewpoint changes.
> Given the additional challenges involved, such as NVS artifacts and noisy correspondance, obtaining high-fidelity pseudo-paired data is non-trivial, and we leave a thorough investigation for future work.
>
> > **Q3.** Correlation between frozen linear probe NMSE and end-to-end VLA fine-tuning performance, and alternative intrinsic metrics.
>
> We agree that NMSE and MI do not guarantee final VLA performance, as other factors such as the action decoder design [4].
> We provide NMSE and MI as diagnostic measures when we pretrain VLA with LAM, but we note that these are **not sufficient conditions** for VLA performance (see also **W1.**, Reviewer SdiK).
> For alternative metrics, bidirectional linear probing that measures NMSE in both the $\text{latent action} \rightarrow \text{action}$ and $\text{action} \rightarrow \text{latent action}$ directions could serve as a more informative proxy, as discussed in **W1.** with Reviewer Y3ps.
>
>
> > **Q4-1.** Per-position code distributions?
>
> We thank the reviewer for this careful suggestion.
> We additionally provide per-position entropy statistics (in bits) below.
>
> | Stat. | MVP-LAM | UniVLA |
> |---|---:|---:|
> | Avg. | $12.81\pm0.03$ | $8.24\pm0.01$ |
> | Max. | $13.26\pm0.02$ | $8.64\pm0.05$ |
> | Min. | $11.89\pm0.04$ | $7.77\pm0.08$ |
>
> These results indicate that all codebook positions are actively utilized.
>
> > **Q4-2.** Increasing codebook size or code length improve OOD generalization?
>
> We believe so, as increasing codebook size and code length improves $\text{latent action} \rightarrow \text{action}$ NMSE (refer to discussion in **Q.** with Reviewer Sdik), though this may come at the cost of latent action minimality.
>
> [1] Sermanet, P., Lynch, C., Hsu, J., & Levine, S. (2017). Time-contrastive networks: Self-supervised learning from multi-view observation. arXiv preprint arXiv:1704.06888.
>
> [2] Tian, S., Wulfe, B., Sargent, K., Liu, K., Zakharov, S., Guizilini, V. C., & Wu, J. (2024). View-invariant policy learning via zero-shot novel view synthesis. In 8th Annual Conference on Robot Learning. https://openreview.net/forum?id=tqsQGrmVEu.
>
> [3] Luo, M., Xue, Z., Dimakis, A., & Grauman, K. (2025). Viewpoint Rosetta Stone: Unlocking unpaired ego-exo videos for view-invariant representation learning. In Proceedings of the IEEE/CVF Conference on Computer Vision and Pattern Recognition.
>
> [4] Bu, Q., Yang, Y., Cai, J., Gao, S., Ren, G., Yao, M., Luo, P., & Li, H. (2025). UniVLA: Learning to act anywhere with task-centric latent actions. arXiv. https://arxiv.org/abs/2505.06111

---

> > ### Author Rebuttal · Reviewer_ngzd · 2026-04-04
> >
> > Thank you for your detailed and thoughtful rebuttal. The responses have addressed my main concerns. I maintain my positive assessment of the paper.

---

> > > ### Author Response · Authors · 2026-04-06
> > >
> > > Thank you for acknowledging our rebuttal. We are glad that our responses have adequately addressed your concerns.

---

### Official Review · Reviewer_Y3ps · 2026-03-15

**Soundness:** 3
**Presentation:** 3
**Significance:** 3
**Originality:** 3
**Overall Recommendation:** 5
**Confidence:** 4

**Summary:**

The authors propose to learn latent actions that are consistent across camera viewpoints. By leveraging self- and cross-viewpoint prediction losses, the model is able to learn latent actions that are more atomic and can transfer across positions. Action decoding experiments are then performed to evaluate the quality of the latent actions and VLA experiments to measure performance in real settings, demonstrating improved performance over existing methods.

**Compliance With Llm Reviewing Policy:**

Affirmed.

**Final Justification:**

My initial main concern regarding action minimality has been addressed by the author's rebuttal.

The new experiments on predicting latent actions from real actions show that the model performs decently well at this (hard) task. The newly shared qualitative analyses are also appreciated, particularly with the Ego-Exo data, where we can see that the model focuses on the hands in these very different viewpoints.

I have thus increased my score compared to my initial assessment, and recommend acceptance for this work.

**Key Questions For Authors:**

Beyond the points raised in the weaknesses there is one point I would like to discuss.

As discussed in the weaknesses, a lot of the evaluations do not exactly measure whether or not the learned latent actions are truly “minimal”, in the sense that they would be completely viewpoint invariant. However, trying to go to this goal leads to simpler latent actions which do not need to encode as much information about the viewpoint. This means that the limited capacity of the discrete latent actions can be used for more fine grained motion for example. Do you think that this plays a role in practice and may it be a key factor influencing performance in the VLA setup ?

**Limitations:**

Yes

**Strengths And Weaknesses:**

**Strengths:**
- The method is intuitive, sensible and well explained
- Experiments are not only performed with VLAs but the latent actions are studied by themselves as well, to ensure their quality
- The latent actions lead to a competitive VLA model, outperforming existing methods on benchmarks such as SIMPLER or LIBERO-long.

**Weaknesses:**
- A benefit of cross-viewpoint latent actions would be that they encode “move left by X amount” and not “move from pixel X to pixel Y” for example. However the definition of action centric latent action does not make a difference between the two. Assuming the former is denoted as X, the latter as Z and the real action as Y, we have $I(X,Y) = I(Z,Y)$. This is reflected in the linear probing evaluation where predicting the real action from a latent action also does not separate these cases, and the desirable minimality provided by cross-viewpoint latent actions. Performing the regression other way around: real action -> latent action, is a much harder task but is where we can expect the most benefit from the cross-viewpoint nature of the model. This would allow for a clearer evaluation, and potentially be a way to highlight the strength of the model more
- Table 4: It would be beneficial to also show the combination with only Robot and no $L_{cross}$, to avoid having multiple components always being changed. This would help highlight the contribution of the human data alone, but is a minor point.
- The evaluation of the robustness to viewpoint perturbation in section 4.4 is appreciated but could be supplemented by qualitative analyses. Some of the training data have very diverse viewpoints (e.g. EgoExo4d) and the work would benefit from visualizing the latent action transferability in this case, beyond the few samples in Figure 7. The increase in MSE shown in Table 3 when using the viewpoint-perturbed transition (which are small perturbations from what I can see in Figure 8) is quite high, and this qualitative analysis could help provide a better understanding of the learned latent action.

**Minor points:**
- Equation 3. $S_t$ and $V_t$ are not introduced. I assume this refers to two viewpoints but would need to be clarified

---

> ### Author Rebuttal · Authors · 2026-03-30
>
> We thank the reviewer for the thoughtful and positive review.
> We address the main concerns below.
>
> > **W1.** The definition of action-centric latent action does not make a difference between the two.
>
> We agree that our current definition and the linear probe primarily measure action informativeness, not strict minimality.
> We also agree with the reviewer that, in principle, a latent action encoding an abstract motion and one encoding a view-dependent pixel-space change may not be distinguishable by MI and NMSE if both are equally predictive of the real action.
>
> However, this equivalence does not always hold in practice, *since pixel-space variation in videos is not guaranteed to be induced purely by the agent's action* and may also include viewpoint-specific or other exogenous changes.
> In that sense, the current evaluation is still informative about action-alignment.
>
> Following the reviewer's suggestion, we additionally performed the reverse probe, namely $\text{action}\rightarrow\text{latent action}$ and report NMSE (lower is better).
>
> | Dataset | MVP-LAM | UniVLA |
> |---|---:|---:|
> | Bridge V2 | $\mathbf{0.792\pm0.01}$| $0.875\pm0.01$ |
> | LIBERO-Long | $\mathbf{0.631\pm0.01}$ | $0.650\pm0.01$ |
> | LIBERO-Goal | $0.643\pm0.01$ | $\mathbf{0.587\pm0.01}$ |
>
> These results show a trend similar to the $\text{latent action}\rightarrow\text{action}$ probe and this implies that MVP-LAM achieves action-centricity with minimality compared to UniVLA.
> We will add this analysis and discuss it explicitly in the revision as a more direct test of whether the latent action is determined by the action itself, rather than by action plus viewpoint-specific nuisance factors.
>
> > **W2.** Table 4. It would be beneficial to also show the combination with only Robot and no $\mathcal{L}_{\text{cross}}$, to avoid having multiple components always being changed.
>
> We note that Bu et al. [1] already empirically demonstrates the benefit of including human data.
> However, we acknowledge that this ablation would provide a cleaner separation, and we will include it if the training budget allows.
>
> > **W3.** The evaluation of the robustness to viewpoint perturbation in Section 4.4 could be supplemented by qualitative analyses.
>
> We agree and have prepared additional qualitative analyses, available at [[this link]](https://q7m4p2x9a1n8.s3.ap-southeast-2.amazonaws.com/f8c1d2.pdf).
> Since our decoder predicts in the DINOv2 feature space rather than in pixel space, direct pixel-space reconstruction is not available.
> Instead, we provide attention maps.
> Figure S1 shows that, under moderate viewpoint perturbation (top two rows), MVP-LAM remains more focused on manipulation-relevant regions than UniVLA.
> Under larger perturbations (bottom row), both LAMs place high attention on irrelevant regions.
> Consequently, the inferred latent action fails to capture the motion needed to explain the action-induced next observation, which we believe leads to the larger decoder prediction error reported in Table 3.
> We will include this discussion more clearly in the paper.
>
> > **Q.** Do you think that minimality — by reducing viewpoint-specific encoding — leads to more efficient use of the limited discrete latent capacity for fine-grained motion, and may this be a key factor influencing performance in the VLA setup?
>
> Yes, we believe it is an important practical mechanism.
> Cross-viewpoint reconstruction encourages the latent action to encode the same transition across viewpoints through the decoder.
> As discussed at the end of Section 3.3 (lines 209–219), cross-viewpoint reconstruction discourages the latent action from encoding viewpoint-specific cues, which leads to more efficient use of the limited discrete latent capacity.
> While directly enforcing and verifying that the remaining capacity encodes fine-grained motion information is important future work, we found that simply filtering out viewpoint-specific effects induces more action-centric latent actions efficiently and is likely one of the key factors behind the improved downstream VLA performance.
>
> > **Minor.** Equation 3. $S_t$ and $V_t$ are not introduced. I assume this refers to two viewpoints but would need to be clarified.
>
> We apologize for the unclear notation.
> As defined at the beginning of Section 3.2 (line 187), $S_t$ denotes the state and $V_t$ denotes the viewpoint.
> We will further clarify the meaning of each symbol in the revision.
>
> [1] Bu, Q., Yang, Y., Cai, J., Gao, S., Ren, G., Yao, M., Luo, P., & Li, H. (2025). UniVLA: Learning to act anywhere with task-centric latent actions. arXiv. https://arxiv.org/abs/2505.06111

---

> > ### Author Rebuttal · Reviewer_Y3ps · 2026-04-01
> >
> > Thank you for the detailed answer, my concerns have been addressed.
> >
> > The new experiments on predicting $\text{action}\rightarrow\text{latent action}$ show that the model performs decently well at this (hard) task. The fact that UniVLA achieves similar performance is not an issue at all, as it is the result of a significantly more complex pipeline with more annotations.
> >
> > The newly shared qualitative analyses are also appreciated, particularly with the Ego-Exo data, where we can see that the model focuses on he hands in these very different viewpoints.
> >
> > I would highly encourage the authors to add all of these new analyses in the revised manuscript.
> >
> > My score will be adjusted accordingly.

---

> > > ### Author Response · Authors · 2026-04-02
> > >
> > > We sincerely thank the reviewer for the constructive feedback and for recognizing the value of our additional experiments.
> > > We will incorporate all the suggested analyses, including the $\text{latent action} \rightarrow \text{action}$ prediction results and qualitative visualizations with egoexo4D, into the revised manuscript.

---

### Decision · Program_Chairs · 2026-04-30

**Decision:**

Accept (regular)

**Comment:**

## Summary
The paper proposes Multi-ViewPoint Latent Action Model (MVP-LAM) for learning discrete latent actions from time-synchronized multi-view videos.

## Ratings
The paper initially receives 1 accept, 2 weak accept and 1 weak reject ratings

## Reasons to accept the paper
* The method is intuitive, sensible and well explained
* Strong experimental results, outperforming existing methods on various benchmarks.

## Reasons to reject the paper
* Experiments are limited to simulation environments, without real-robot validation.

## Discussion and Decision
AC reads all reviews from reviewers and the discussion between authors and reviewers. AC found the merit outweighs the remaining concerns (lacking real-world robot validation), thus recommending to accept the paper. AC encourages the author(s) to incorporate rebuttal into their final camera-ready version.